# E2Former: An Efficient and Equivariant Transformer with Linear-Scaling Tensor Products

**Yunyang Li**[1†], **Lin Huang**[2† ‡*], **Zhihao Ding**[3] , **Chu Wang** [4], **Xinran Wei** [5‡],
**Han Yang**[6], **Zun Wang**[7‡], **Chang Liu**[5‡], **Yu Shi**[5‡], **Peiran Jin** [5‡],
**Tao Qin**[5‡], **Mark Gerstein**[1*], **Jia Zhang**[2*‡]

[1]Yale University, [2]Ubiquant, [3]PolyU, [4]HUST, [5]ZGC Academy, [6]Microsoft Research, [7]Shanghai AI Lab
`https://github.com/liyy2/E2Former`

## Abstract

Equivariant Graph Neural Networks (EGNNs) have demonstrated significant success in modeling microscale systems, including those in chemistry, biology and materials science. However, EGNNs face substantial computational challenges due to the high cost of constructing edge features via spherical tensor products, making them almost impractical for large-scale systems. To address this limitation, we introduce E2Former, an equivariant and efficient transformer architecture that incorporates a Wigner $6j$ convolution (Wigner $6j$ Conv). By shifting the computational burden from edges to nodes, Wigner $6j$ Conv reduces the complexity from $O(|\mathcal{E}|)$ to $O(|\mathcal{V}|)$ while preserving both the model's expressive power and rotational equivariance. We show that this approach achieves a 7x–30x speedup compared to conventional $SO(3)$ convolutions. Furthermore, our empirical results demonstrate that the derived E2Former mitigates the computational challenges of existing approaches without compromising the ability to capture detailed geometric information. This development could suggest a promising direction for scalable molecular modeling.

## 1 Introduction

Molecular simulations underpin critical computational tasks across chemistry [31, 36, 38, 39], biology [9], and materials science [64], facilitating detailed exploration of microscopic processes. Although quantum mechanical approaches such as Density Functional Theory (DFT) provide highly accurate predictions [30, 37], their computational complexity scales poorly with system size [56], thus limiting practical applicability to small-scale problems. Machine Learning (ML) techniques have emerged as promising alternatives, balancing computational efficiency and accuracy [5, 4, 16]. ML-based models, particularly Equivariant Graph Neural Networks (EGNNs), significantly reduce simulation times, enabling molecular property predictions and dynamic simulations within practical computational budgets [53, 26, 25, 7, 23, 41]. EGNN architectures explicitly encode symmetry constraints—such as rotational and reflectional equivariances—through graph-based atomic representations. This symmetry-awareness leads to strong inductive biases and improved sample efficiency. EGNNs have evolved from rotationally invariant embedding methods like SchNet [53] to schemes incorporating bond and dihedral angles [26, 25], scalarization techniques [52, 63], and spherical tensor-product frameworks such as E(3) and SE(3)-Transformers [57, 28, 23, 41]. Recent refinements, including Gaunt Tensor Product [44] eSCN convolutions [47, 42], primarily focus on enhancing computational efficiency.

---

[†]Co-first authors.

[‡]Part of this work was completed while the authors were at Microsoft Research.

[*]Corresponding authors. `huang_6385@outlook.com`, `pi@gersteinlab.org`, `jialrs.z@gmail.com`
[*]Note: JZ and LH (Ubiquant) thank ScitiX for computing power and training infrastructure. MG is supported by the ALW professorship fund.

39th Conference on Neural Information Processing Systems (NeurIPS 2025).

In this work, we specifically focus on spherical-equivariant EGNN architectures [57, 23, 41], which leverage spherical harmonics and Clebsch–Gordan tensor products. These models—commonly referred to as *spherical EGNNs*—have demonstrated state-of-the-art accuracy, especially for periodic systems where symmetry constraints are critical [59, 10]. By encoding higher-order geometric correlations through irreducible representations (irreps) with angular momentum $L > 1$, spherical EGNNs offer expressive, data-efficient models capable of capturing complex geometric interactions [57, 55]. Unfortunately, these gains come at a computational cost. The use of spherical tensor products for feature construction incurs complexity driven by two factors: (i) the number of tensor products required, which scales with the number of edges $|\mathcal{E}|$ in the molecular graph, and (ii) the computational cost of each tensor product, which grows with the angular momentum cutoff $L$. Together, these lead to runtime costs of $O(|\mathcal{E}|L^6)$ or $O(|\mathcal{E}|L^3)$ when implemented with the sparse eSCN convolution. This scaling presents a significant bottleneck, limiting the use of spherical EGNNs to small- or medium-scale systems, despite their improved performance in principle. While recent spherical-scalarization methods [52, 63, 2] offer efficient alternatives by bypassing tensor products, they sacrifice theoretical completeness [18]. Tensor-product formulations, in contrast, preserve the full space of equivariant functions between irreps. This trade-off motivates our effort to retain the expressive power of tensor products while eliminating their prohibitive complexity.

Here, we introduce the Wigner $6j$ convolution (Wigner $6j$ Conv, Figure 1), a spherical-equivariant method that uses Wigner $6j$ symbols [40, 45, 20], *provably* reducing tensor product complexity to $O(|\mathcal{V}|)$ while maintaining the *exact expressive power* and rotational equivariance.

> **Contributions.** Our contributions can be summarized as follows: (1) We introduce the Wigner $6j$ convolution, a spherical-equivariant technique that reduces the computational complexity from $O(|\mathcal{E}|)$ to $O(|\mathcal{V}|)$, enabling the modeling of larger molecular systems without compromising the network's expressive power or symmetry properties. As shown in Figure 2(b), our model demonstrates better scaling behavior than the $SO(3)$ convolution, achieving 7x to 30x speed-up given the sparsity of the molecular graph. (2) We propose E2Former, an equivariant and efficient Transformer architecture specifically designed for scalable molecular modeling. E2Former leverages the Wigner $6j$ convolution to maintain rotational equivariance while significantly enhancing computational efficiency. (3) Extensive experiments on benchmark datasets like OC20, OC22, and SPICE show that E2Former achieves competitive accuracy in predicting molecular energies and forces, while offering improved efficiency and scalability over existing spherical-equivariant methods. (4) Finally, we pre-trained E2Former on a large-scale dataset and evaluated its performance in molecular dynamics simulations, where it achieves high accuracy with faster speed, outperforming state-of-the-art empirical potential methods and EGNNs. These results suggest its potential to advance large-scale molecular simulations and to serve as a foundational model for machine learning force fields.

## 2   Background and Preliminaries

**Notation.** Throughout this paper, we use $\ell$ and $m$ to denote angular momentum quantum numbers associated with spherical harmonics $Y_m^{(\ell)}$, where $\ell \geq 0$ and $-\ell \leq m \leq \ell$. All spherical harmonics are considered real-valued functions on $\mathbb{R}^3$. Positions of nodes in $\mathbb{R}^3$ are represented as $\mathbf{r}$, while node-level irreducible features are denoted $\mathbf{h}_i \in \mathbb{R}^{s \times c}$, where $s$ represents the spherical dimension and $c$ the feature dimension per spherical component (i.e. number of channels). The operation $[...]^{(\ell)}$ is the *projection operation* which extracts only the $\ell$-th order irreducible component from a representation or tensor product. Clebch-Gorden Tensor products of irreps are symbolized by $\otimes$ (*without* any superscripts), and its Wigner $6j$ counterpart is denoted by $\otimes^{6j}$.

In this section, we establish the mathematical foundations necessary for constructing the Wigner-$6j$ Convolution. These include real-space *solid spherical harmonics*, tensor products of irreducible representations (irreps), and Wigner $6j$ *recoupling* theory. We commence by defining the solid spherical harmonics in real basis, which is commonly used in modern ML applications:

**Definition 2.1** (Solid Spherical Harmonics in Real Basis). Let $\mathbf{r} = (x, y, z) \in \mathbb{R}^3$, $r = \|\mathbf{r}\| = \sqrt{x^2 + y^2 + z^2}$, and $(r, \theta, \phi)$ be the spherical coordinates with:

$$\theta = \arccos\left(\frac{z}{r}\right), \qquad \phi = \text{atan2}(y, x).$$

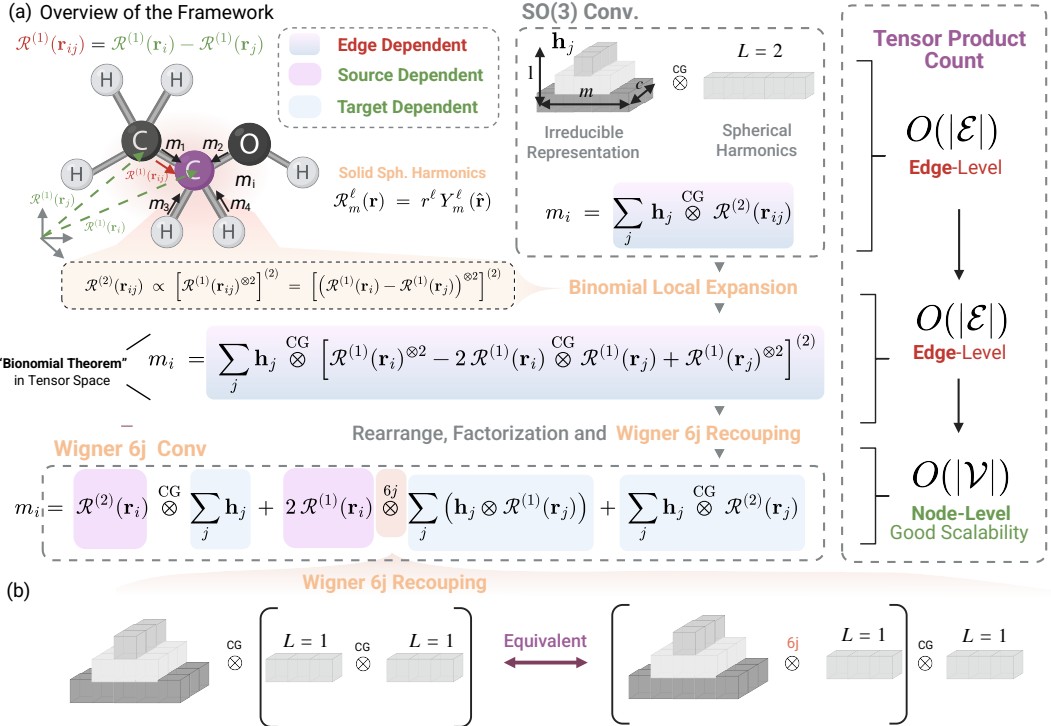

Figure 1: (**a**) Overview of the Proposed Approach. Rather than performing tensor products over edges by combining node features and distances, E2Former leverages two key concepts: *binomial local expansion* and *Wigner* $6j$ *recoupling*. The former represents edge directions in terms of node positions, while the latter reorders the sequence of tensor product operations. Together, the computational complexity of the tensor product is reduced from $O(|\mathcal{E}|)$ to $O(|\mathcal{V}|)$. $\otimes$ denotes the Clebsch-Gorden tensor product, and $\otimes^{6j}$ denotes the CG tensor product where each path is parameterized by a weight governed by the Wigner-$6j$ coefficients. (**b**) Illustration of two equivalent ways to couple the tensor product of three representations: sequentially coupling two tensors before the third (left) or reordering the coupling sequence (right), with equivalence established via the Wigner $6j$ recoupling.

The *(regular) solid spherical harmonics* are homogeneous harmonic polynomials of degree $\ell$ defined by:

$$\mathcal{R}_m^{(\ell)}(\mathbf{r}) \;=\; r^\ell \, Y_m^{(\ell)}(\hat{\mathbf{r}}), \qquad \hat{\mathbf{r}} = \mathbf{r}/r, \quad \ell \geq 0, \; -\ell \leq m \leq \ell,$$

where $Y_m^{(\ell)}$ are real spherical harmonics on $S^2$. Equivalently, in $(r, \theta, \phi)$,

$$\mathcal{R}_m^{(\ell)}(r, \theta, \phi) = \begin{cases} k_m^{(\ell)} \, r^\ell \, P_m^{(\ell)}(\cos\theta) \, \cos(m\phi), & m > 0, \\ k_0^{(\ell)} \, r^\ell \, P_0^{(\ell)}(\cos\theta), & m = 0, \\ k_{|m|}^{(\ell)} \, r^\ell \, P_{|m|}^{(\ell)}(\cos\theta) \, \sin(|m|\phi), & m < 0, \end{cases}$$

with $P_{|m|}^{(\ell)}$ the associated Legendre polynomials and $k_m^{(\ell)}$ normalization constants (chosen per the convention used in this work). For example, in e3nn implementation, the real spherical harmonics for $\ell = 2$ take the following form in Cartesian coordinates: $\mathcal{R}_{-2}^{(2)}(x, y, z) = xy, \mathcal{R}_{-1}^{(2)}(x, y, z) = yz, \mathcal{R}_0^{(2)}(x, y, z) = 3z^2 - (x^2 + y^2 + z^2), \mathcal{R}_1^{(2)}(x, y, z) = xz, \mathcal{R}_2^{(2)}(x, y, z) = x^2 - y^2$. Under a rotation $g \in \mathrm{SO}(3)$, these functions transform according to: $\mathcal{R}_m^{(\ell)}(\mathbf{r}) \mapsto \sum_{m'=-\ell}^{\ell} D_{m,m'}^{(\ell)}(g) \, \mathcal{R}_{m'}^{(\ell)}(\mathbf{r})$, where $D_{m,m'}^{(\ell)}(g)$ are the Wigner $D$-matrices. This transformation rule ensures that spherical harmonics of fixed degree $\ell$ transform properly under the action of $\mathrm{SO}(3)$. One useful property of solid spherical harmonics that will come in useful later is $\mathcal{R}^{(1)}(\mathbf{r}_{ij}) = \mathcal{R}^{(1)}(\mathbf{r}_i) - \mathcal{R}^{(1)}(\mathbf{r}_j)$. It is also worth noting that this equality holds universally if and only if $\ell = 1$.

Next, we introduce the behavior of irreducible representations (irreps). A key principle is that the tensor product of two irreps is generally reducible, meaning it decomposes into a direct sum of other

irreps. This decomposition mechanism is precisely what will allow us to relate the general irrep $\mathcal{R}^{(\ell)}(\cdot)$ back to $\mathcal{R}^{(1)}(\cdot)$.

**Definition 2.2** (Tensor Products of Irreps). Let $U^{(\ell_1)}$ and $U^{(\ell_2)}$ be irreducible representations (irreps) of SO(3). Their tensor product $U^{(\ell_1)} \otimes U^{(\ell_2)}$ decomposes into a direct sum of irreps: $U^{(\ell_1)} \otimes U^{(\ell_2)} = \bigoplus_{\ell=|\ell_1-\ell_2|}^{\ell_1+\ell_2} U^{(\ell)}$. The decomposition is governed by the **Clebsch–Gordan coefficients**. Specifically, the tensor product, projected onto a specific irreducible component $U^{(\ell_3)}$, is denoted as:

$$\left[ U^{(\ell_1)} \otimes U^{(\ell_2)} \right]^{(\ell_3)} = \sum_{\ell_1 m_1, \ell_2 m_2} C^{\ell_3 m_3}_{\ell_1 m_1, \ell_2 m_2} U^{(\ell_1)}_{m_1} U^{(\ell_2)}_{m_2}. \tag{2.1}$$

**Commutativity of the Clebsch–Gordan Tensor Product.** The tensor product of two irreducible representations (irreps) $U^{(a)}$ and $U^{(b)}$ of SO(3) is not strictly commutative as a bilinear operation on vector spaces: $U^{(a)} \otimes U^{(b)}$ is not identical to $U^{(b)} \otimes U^{(a)}$. Nonetheless, this operation is *effectively* commutative at the level of irreducible decompositions. Interchanging the order of the factors does not change the set of irreps that appear, although it permutes the corresponding CG coefficients. In particular, we have:

$$U^{(a)} \otimes U^{(b)} \cong U^{(b)} \otimes U^{(a)} \cong \bigoplus_{j=|a-b|}^{a+b} U^{(j)}. \tag{2.2}$$

**Associativity of the Clebsch–Gordan Tensor Product.** For irreducible representations (irreps) of SO(3), the tensor product is associative up to a canonical isomorphism. Specifically, for any three irreps $U^{(a)}$, $U^{(b)}$, and $U^{(c)}$, the following holds:

$$(U^{(a)} \otimes U^{(b)}) \otimes U^{(c)} \cong U^{(a)} \otimes (U^{(b)} \otimes U^{(c)}).$$

While the set of resulting irreps is independent of the association order, the CG coefficients that appear in the decomposition do depend on the chosen coupling scheme. Transitions between different coupling orders are governed by Wigner $6j$ symbols, which express changes of basis without modifying the underlying irreducible content.

**Definition 2.3** (Wigner $6j$ Symbol). For three irreps $U^{(a)}$, $U^{(b)}$, and $U^{(c)}$ of SO(3), one can couple them either as $U^{(a)} \otimes (U^{(b)} \otimes U^{(c)})$ or as $(U^{(a)} \otimes U^{(b)}) \otimes U^{(c)}$. The Wigner $6j$ symbol $\begin{Bmatrix} a & b & d \\ c & \ell & j \end{Bmatrix}$ relates these two coupling schemes through the identity:

$$\left[ U^{(a)} \otimes \left[ U^{(b)} \otimes U^{(c)} \right]^{(j)} \right]^{(\ell)} = \sum_d (-1)^{a+b+c+d} \sqrt{(2d+1)(2j+1)} \begin{Bmatrix} a & b & d \\ c & \ell & j \end{Bmatrix} \left[ U^{(a)} \otimes U^{(b)} \right]^{(d)} \otimes U^{(c)}. \tag{2.3}$$

To simplify notation, we abstract the recoupling process as follows:

$$U^{(a)} \otimes (U^{(b)} \otimes U^{(c)}) = (U^{(a)} \otimes U^{(b)}) \otimes^{6j} U^{(c)}, \tag{2.4}$$

where $\otimes^{6j}$ denotes a CG tensor product accompanied by a re-indexing via Wigner $6j$ coefficients.

## 3  Wigner $6j$ Convolution

In this section, we introduce the SO(3)-Equivariant Node convolution and demonstrate how Wigner $6j$ recoupling facilitates an efficient node-wise computation.

**Definition 3.1** (SO(3)-Equivariant Node Convolution). Let $\mathbf{h}_i \in \mathbb{R}^{s \times c}$ denote the irreducible feature tensor of node $i$, where $s$ indexes the irreducible representation (irrep) type and $c$ indexes the channels within each irrep. Let $\mathcal{R}^{(\ell)}(\mathbf{r}_{ij})$ denote the degree-$\ell$ spherical harmonic evaluated at the relative direction. The SO(3)-equivariant node convolution is via the CG tensor product between the source irreps and the spherical harmonics: $\mathbf{h}_i := \sum_{j \in \mathcal{N}(i)} \mathbf{h}_j \otimes \mathcal{R}^{(\ell)}(\mathbf{r}_{ij})$.

We clarify that our formulation of the SO(3) convolution employs $\mathcal{R}(\cdot)$ rather than $Y(\cdot)$. The two formulations are related through a normalization factor. To realize the $Y(\cdot)$-based variant, this normalization factor can be absorbed into the attention coefficients, as detailed in Alg. 1.

**Wigner** $6j$ **convolution.** Given the SO(3) convolution, we aim to demonstrate that the operation admits a node-wise factorization via Wigner $6j$ symbols. In particular, we show that the SO(3) convolution can be expressed as:

$$\mathbf{h}_i = \sum_{j \in \mathcal{N}(i)} \underbrace{\left(\mathbf{h}_j \otimes \mathcal{R}^{(\ell)}(\mathbf{r}_{ij})\right)}_{\text{ij-dependent}} = \sum_{u=0}^{\ell}(-1)^{\ell-u}\binom{\ell}{u}\underbrace{\left(\mathcal{R}^{(u)}(\mathbf{r}_i)\right)}_{\text{i-dependent}} \otimes^{6j} \left(\sum_{j \in \mathcal{N}(i)} \underbrace{\mathbf{h}_j \otimes \left(\mathcal{R}^{(\ell-u)}(\mathbf{r}_j)\right)}_{\text{j-dependent}}\right).$$

The blue–boxed factors $\mathcal{R}^{(u)}(\vec{r}_i)$ aggregate all node-$i$–specific terms, whereas the red–boxed factors $\mathbf{h}_j, \mathcal{R}^{(\ell-u)}(\vec{r}_j)$ isolate the node-$j$ contribution. This separation removes explicit edge dependencies, resulting in the number of tensor products in the network scaling with $O(|V|)$. To build further intuition, we draw an analogy to factorization techniques in kernelized attention mechanisms [13], which achieve linear scaling by decoupling query-key interactions.

To establish this result, we introduce the concept of the *Binomial Local Expansion*. The expansion is based on the key insight that any term $\mathcal{R}^{(\ell)}(\cdot)$ of arbitrary order $\ell$ can be expressed through iterative tensor products of the first-order term, $\mathcal{R}^{(1)}(\cdot)$. This effectively reduces the problem to the first-order case, where we can apply the previously introduced relation $\mathcal{R}^{(1)}(\mathbf{r}_{ij}) = \mathcal{R}^{(1)}(\mathbf{r}_i) - \mathcal{R}^{(1)}(\mathbf{r}_j)$ to factor the edge-dependent expression into node-local terms.

**Theorem 3.2** (Bionomial Local Expansion). *Let $\ell = u \geq 1$. Every $\ell = u$ spherical harmonic $\mathcal{R}^{(l)}(\mathbf{r}_{ij})$ can be expressed as an irreducible subspace of the $u$-fold tensor product $(\mathcal{R}^{(1)}(\mathbf{r}_{ij}))^{\otimes u}$. When expanded in terms of node-local terms, this satisfies:*

$$\mathcal{R}^{(\ell)}(\mathbf{r}_{ij}) = \sum_{u=0}^{\ell}(-1)^{\ell-u}\binom{\ell}{u}\left[\left(\mathcal{R}^{(u)}(\mathbf{r}_i)\right) \otimes \left(\mathcal{R}^{(\ell-u)}(\mathbf{r}_j)\right)\right]^{(\ell)},$$

*Proof Sketch.* This spherical harmonic $\mathcal{R}^{(\ell)}(\mathbf{r}_{ij})$ could be constructed by projecting the $\ell$-fold tensor product of the first-order harmonic $\mathcal{R}^{(1)}(\mathbf{r}_{ij})$ onto the subspace transforming as the irreducible representation (irrep) $\ell$ of SO(3). Recall that the projection operator is denoted by $[...]^{(\ell)}$ and using the identity $\mathcal{R}^{(1)}(\mathbf{r}_{ij}) = \mathcal{R}^{(1)}(\mathbf{r}_i) - \mathcal{R}^{(1)}(\mathbf{r}_j)$, the objective could be rewritten as $\mathcal{R}^{(\ell)}(\mathbf{r}_{ij}) = [(\mathcal{R}^{(1)}(\mathbf{r}_i) - \mathcal{R}^{(1)}(\mathbf{r}_j))^{\otimes\ell}]^{(\ell)}$.

We begin by expanding the tensor power $(\mathcal{R}^{(1)}(\mathbf{r}_i) - \mathcal{R}^{(1)}(\mathbf{r}_j))^{\otimes\ell}$, which produces a sum of $2^\ell$ tensor products. Each term corresponds to an ordered sequence $P \in \{i, j\}^\ell$, where each factor is either $\mathcal{R}^{(1)}(\mathbf{r}_i)$ or $\mathcal{R}^{(1)}(\mathbf{r}_j)$. Denote the corresponding tensor product as $T_P$. For example, if $P = (i, j, i)$, then $T_P = \mathcal{R}^{(1)}(\mathbf{r}_i) \otimes \mathcal{R}^{(1)}(\mathbf{r}_j) \otimes \mathcal{R}^{(1)}(\mathbf{r}_i)$. Initially, each ordering $(*, *, \cdots, *)$ defines a distinct term. Later, we will show that the projection operator renders the result invariant to the ordering. We write the full expansion as: $(\mathcal{R}^{(1)}(\mathbf{r}_i) - \mathcal{R}^{(1)}(\mathbf{r}_j))^{\otimes\ell} = \sum_{u=0}^{\ell}(-1)^{\ell-u}\sum_{P\in\Pi_u} T_P$, where $\Pi_u$ denotes the set of orderings containing exactly $u$ factors of $\mathcal{R}^{(1)}(\mathbf{r}_i)$ and $\ell - u$ factors of $\mathcal{R}^{(1)}(\mathbf{r}_j)$. Applying the linear projection operator $[...]^{(\ell)}$ to this sum distributes the operator yields $[(\mathcal{R}^{(1)}(\mathbf{r}_i) - \mathcal{R}^{(1)}(\mathbf{r}_j))^{\otimes\ell}]^{(\ell)} = \sum_{u=0}^{\ell}(-1)^{\ell-u}\sum_{P\in\Pi_u}[T_P]^{(\ell)}$.

The key insight comes from angular momentum coupling theory. Combining $\ell$ systems with angular momentum 1 yields components with total angular momentum ranging up to $\ell$. The subspace associated with the *highest possible* angular momentum, $L = \ell$, is *unique* and corresponds to the *fully symmetric* combination of the individual factors. The projector $[...]^{(\ell)}$ isolates precisely this unique, symmetric component. As a result, the projected tensor $[T_P]^{(\ell)}$ remains identical for all orderings $P \in \Pi_u$, indicating that the projection depends solely on the multiplicities of the factors $\mathcal{R}^{(1)}(\mathbf{r}_i)$ and $\mathcal{R}^{(1)}(\mathbf{r}_j)$ in $T_P$, rather than their ordering. The inner sum over the $\binom{\ell}{u}$ identical projected terms simplifies. Let $T_{\text{rep}} = (\mathcal{R}^{(1)}(\mathbf{r}_i))^{\otimes u} \otimes (\mathcal{R}^{(1)}(\mathbf{r}_j))^{\otimes(\ell-u)}$ serve as a representative tensor product for the class $\Pi_u$. Then: $\sum_{P\in\Pi_u}[T_P]^{(\ell)} = |\Pi_u|[T_{\text{rep}}]^{(\ell)} = \binom{\ell}{u}[(\mathcal{R}^{(1)}(\mathbf{r}_i))^{\otimes u} \otimes (\mathcal{R}^{(1)}(\mathbf{r}_j))^{\otimes(\ell-u)}]^{(\ell)}$.

Substituting this simplification back into the expression for the projected tensor power yields: $[(\mathcal{R}^{(1)}(\mathbf{r}_i) - \mathcal{R}^{(1)}(\mathbf{r}_j))^{\otimes\ell}]^{(\ell)} = \sum_{u=0}^{\ell}(-1)^{\ell-u}\binom{\ell}{u}[(\mathcal{R}^{(1)}(\mathbf{r}_i))^{\otimes u} \otimes (\mathcal{R}^{(1)}(\mathbf{r}_j))^{\otimes(\ell-u)}]^{(\ell)}$.

$\square$

**Theorem 3.3** (Node-Based Factorization via Wigner $6j$). SO(3) *convolutions admit a factorization that separates the dependence on the central node $i$ from the aggregation over neighbors $j$, yielding the form:*

$$\sum_{j \in \mathcal{N}(i)} \mathbf{h}_j \otimes \mathcal{R}_m^{(\ell)}(\mathbf{r}_{ij}) = \sum_{u=0}^{\ell} (-1)^{\ell-u} \binom{\ell}{u} \left( \mathcal{R}^{(u)}(\mathbf{r}_i) \right) \otimes^{6j} \left( \sum_{j \in \mathcal{N}(i)} \mathbf{h}_j \otimes \left( \mathcal{R}^{(\ell-u)}(\mathbf{r}_j) \right) \right),$$

*where $\otimes^{6j}$ denotes a CG tensor product where the path weight is parameterized by the corresponding Wigner $6j$ coefficients.*

*Proof Sketch.* We begin by substituting the spherical harmonic $\mathcal{R}_m^{(\ell)}(\mathbf{r}_{ij})$ using the binomial local expansion from into the original SO(3)-equivariant convolution expression, we obtain:

$$\sum_{j \in \mathcal{N}(i)} \mathbf{h}_j \otimes \sum_{u=0}^{\ell} (-1)^{\ell-u} \binom{\ell}{u} \left[ \mathcal{R}^{(u)}(\mathbf{r}_i) \otimes \mathcal{R}^{(\ell-u)}(\mathbf{r}_j) \right]^{(\ell)}. \tag{3.1}$$

By linearity of the tensor product, this expression becomes:

$$\sum_{u=0}^{\ell} (-1)^{\ell-u} \binom{\ell}{u} \sum_{j \in \mathcal{N}(i)} \left( \mathbf{h}_j \otimes \left[ \mathcal{R}^{(u)}(\mathbf{r}_i) \otimes \mathcal{R}^{(\ell-u)}(\mathbf{r}_j) \right]^{(\ell)} \right). \tag{3.2}$$

To reorganize this expression in terms of node-dependent features, we apply Wigner $6j$ recoupling. Letting $A = \mathbf{h}_j, B = \mathcal{R}^{(\ell-u)}(\mathbf{r}_j), C = \mathcal{R}^{(u)}(\mathbf{r}_i)$, The recoupling identity states: $A \otimes (B \otimes C) = (A \otimes B) \otimes^{6j} C$. Since CG tensor products commute effectively, we can swap $B$ and $C$ before applying recoupling. This gives:

$$\mathbf{h}_j \otimes \left( \mathcal{R}^{(u)}(\mathbf{r}_i) \otimes \mathcal{R}^{(\ell-u)}(\mathbf{r}_j) \right) = (\mathbf{h}_j \otimes \mathcal{R}^{(\ell-u)}(\mathbf{r}_j)) \otimes^{6j} \mathcal{R}^{(u)}(\mathbf{r}_i). \tag{3.3}$$

Applying this recoupling within the sum, we arrive at the factorized form as claimed. Note that it is safe to apply the recoupling within a projection operator. This constraint can be implemented by fixing one of the intermediate coupling indices in the Wigner $6j$ symbol to $\ell$. $\qquad\square$

We now formalize the key properties of the resulting Wigner $6j$ convolution. These results are stated in the following lemmas. Proofs are provided in Appendix E and Appendix F, respectively.

**Lemma 3.4** (Equivariance of Wigner $6j$ Convolution). *The Wigner $6j$ convolution operator (denoted as $F$), is equivariant under the Euclidean group $\mathrm{SE}(3)$. That is, for any rigid transformation $g \in \mathrm{SE}(3)$, the output satisfies $\mathcal{F}[g \cdot f] = D(g) \cdot \mathcal{F}[f]$.*

**Lemma 3.5** (Time Complexity of Wigner $6j$ Convolution). *The time complexity of Wigner $6j$ convolution is $O((L^6 C + C^2 L^2)|\mathcal{V}|)$, where $L$ is the degree cutoff and $C$ is the number of channels.*

**Model Architecture.** Based on these, we propose *E2Former*, a modular architecture that alternates between E2Attention and feed-forward layers (Appendix Fig. 5(a); additional architectural details provided in the Appendix H). At its core lies a convolution layer based on the Wigner $6j$ convolution, which serves as a backend kernel to efficiently capture rotational symmetries. We highlight two key design considerations underlying E2Former. First, we observe that the attention computation constitutes only a small fraction of the overall runtime in an attention-based SO(3) convolution (Figure 2 (**a**)). Leveraging this, we integrate the attention mechanism directly into the Wigner $6j$ convolution (Algorithm 1). As a result, while the resulting model *is not strictly linear*, the number of tensor product operations scales linearly with input size, enabling efficient computation. Second, E2Former is not defined solely by its use of Wigner $6j$ convolutions. Rather, it represents a broader architectural principle that combines symmetry-aware design with practical engineering. The efficiency gains afforded by the Wigner $6j$ kernel allow us to reallocate the computational budget toward increased expressivity—e.g., by incorporating deeper layers, wider hidden dimensions, more attention heads, and MACE higher-order interactions [6]. This trade-off between symmetry-driven modeling and architectural scalability is central to the design philosophy of E2Former.

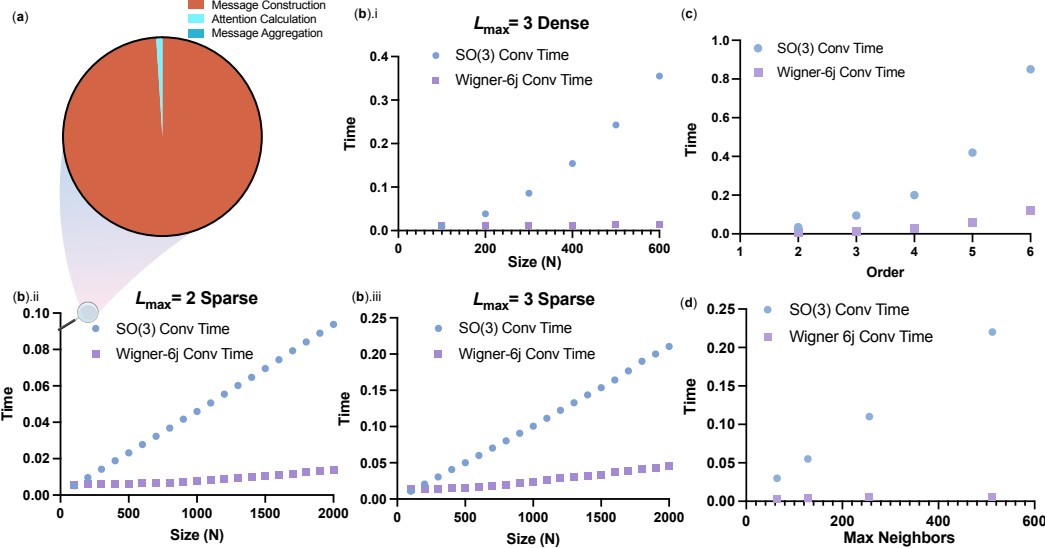

Figure 2: **(a)** Breaking down the runtime of attention-based $SO(3)$ convolutions shows that message construction is the slowest step. Calculating attention and combining messages take much less time. **(b)** We compared the runtime of our Wigner $6j$ convolution (purple squares) against the standard $SO(3)$ convolution (blue circles). Our method was consistently faster across different graph sizes ($N$), maximum angular momenta ($L_{\max}$), and sparsity levels (dense vs. sparse, see subplots b.i-iii). Full experimental details are in Sec. 4.1. **(c)** Runtime on 1000-node graphs as a function of angular momentum cutoff $L$ (up to $L_{\max} = 6$). **(d)** Runtime on 1000-node graphs with fixed $L_{\max} = 3$, varying the maximum number of neighbors from 64 to 512. In **(b–d)**, both methods yield **identical outputs**.

## 4 Results

### 4.1 Scaling Analysis of Wigner $6j$ Conv and $SO(3)$ Conv

Here, we compare the runtime of Wigner $6j$ convolution (purple squares) and $SO(3)$ convolution (blue circles). The two implementations are mathematically equivalent and, by construction, produce *identical outputs* given the same molecular graphs. Specifically, we compare different graph sizes $N$, maximum angular momenta $L_{\max}$, and connectivity patterns. For dense graphs, defined as graphs where every node is connected to all other nodes, at $L_{\max} = 3$ (Fig. 2 (b.i)), the quadratic scaling of $SO(3)$ convolution introduces a noticeable performance gap. Additionally, for sparse graphs, defined here as graphs with $k$-nearest neighbor connectivity ($k = 32$), at $L_{\max} = 2$ and $L_{\max} = 3$ (Figs. 2 (b.ii) and (b.iii)), Wigner $6j$ convolution scales consistently better than the $SO(3)$ convolution. Fig. 2c further shows the impact of increasing the angular momentum cutoff up to $L_{\max} = 6$ on 1000-node graphs, where our method consistently achieves approximately a 7× speed-up over the baseline. Finally, Fig. 2d demonstrates that as the number of neighbors per node increases from 64 to 512 (with $L_{\max} = 3$), the speed-up from Wigner $6j$ convolution becomes even more pronounced.

### 4.2 E2Former Results

We evaluate E2Former which heavily utilizes the Wigner $6j$ convolution on three standard benchmarks—two catalysis datasets (OC20, OC22) and a molecular conformer dataset (SPICE)—and find that it achieves strong accuracy while maintaining computational efficiency.

#### 4.2.1 Performance on the OC20 Dataset

**Dataset Description.** The OC20 dataset [10] comprises 1.2 million DFT relaxations computed using the revised Perdew-Burke-Ernzerhof (RPBE) functional [29]. Each system, averaging 73 atoms, represents an adsorbate molecule on a catalyst surface and is designed for the Structure-to-Energy-and-Forces (S2EF) task. This task involves predicting the system's energy and per-atom forces, with performance evaluated based on the mean absolute error (MAE) of these predictions. Following

[27, 41], we use the **2M** subset for training, and evaluate on the *validation split*. This choice also reflects practical computational constraints, as training on the full dataset requires significant time and resources. All reported results are taken directly from previous publications [27, 41]. We compare two model variants: the 33M-parameter version and the 67M-parameter version. A summary of the results is presented in Table 1. E2Former demonstrates strong performance across all model sizes, with 67M variant achieving results comparable to state-of-the-art methods. Notably, the Small variant (33M parameters) maintains competitive accuracy while offering significant computational advantages.

Table 1: Performance on the OC20-**2M** dataset. Results are reported in Energy (meV) and Force (meV/Å) mean absolute error (MAE). E2Former achieves competitive accuracy and computational efficiency. Approximate training GPU hours are measured on 32G NVIDIA V100 GPUs. The best results are bolded and the second best are highlighted with underline.

| Model | # Params (M) | Training GPU Hours | Inference Speed (samples/sec) | Validation | |
|---|---|---|---|---|---|
| | | | | Energy MAE (meV) | Force MAE (meV/Å) |
| GemNet-dT | **31** | 900 | 50 | 358 | 29.50 |
| GemNet-OC | 38 | 1500 | 38 | 286 | 25.70 |
| SCN | 126 | 3000 | 5 | 279 | 21.90 |
| eSCN | 51 | 2200 | 19 | 283 | 20.50 |
| EquiformerV2 | 85 | 1800 | 19 | 285 | **20.46** |
| E2Former 33M | 33 | **800** | **62** | 275 | 21.90 |
| E2Former 67M | 67 | 1500 | 34 | **270** | 20.50 |

### 4.2.2 Performance on the OC22 Dataset

The OC22 dataset [59] is specifically designed for studying oxide electrocatalysis. In contrast to OC20, OC22 features DFT total energies, which could serve as a general and versatile DFT surrogate, enabling investigations beyond adsorption energies. We train on the OC22 S2EF-Total task and measure energy and force MAE on the S2EF-Total validation splits. Table 2 summarizes our results on the OC22 S2EF task. E2Former achieves competitive energy and force MAEs while enabling rapid training and inference. Notably, it converges in just 1,500 GPU hours—only one-third of the runtime required by the SOTA model.

Table 2: Performance on the OC22 S2EF task. Results are reported for Energy MAE (meV) and Force MAE (meV/Å) under In Distribution (ID) and Out-Of-Distribution (OOD) splits. Approximate training GPU hours are measured on 32G NVIDIA V100 GPUs. The best results are bolded and the second best are highlighted with underline.

| Model | # Params (M) | Training GPU Hours | Energy MAE (meV) | | Force MAE (meV/Å) | |
|---|---|---|---|---|---|---|
| | | | ID | OOD | ID | OOD |
| GemNet-OC | 39 | - | 545 | 1011 | 30.00 | 40.00 |
| EquiformerV2 | 122 | 4500 | **433** | **629** | **22.88** | **30.70** |
| E2Former 67M | 67 | **1500** | 491 | 724 | 25.98 | 36.45 |

### 4.2.3 Performance on the SPICE Dataset

The SPICE dataset [19] comprises small organic molecules and encompasses a diverse array of chemical species with neutral formal charges. The geometries were generated through molecular dynamics simulations using classical force fields, followed by the sampling of various conformations. High-fidelity labeling was achieved at the $\omega$B97M-D3(BJ)/def2-TZVPPD level of calculations. This dataset includes configurations of up to 50 atoms. It was further augmented with larger molecules, ranging from 50 to 90 atoms, derived from the QMugs dataset [32], as well as water clusters obtained from simulations of liquid water. Approximately 85% of the SPICE dataset was used for model training, while 15% was allocated for model testing. We evaluate E2Former 33M on the SPICE dataset and a summary of the results is provided in Table 3.

Table 3: Performance comparison on the SPICE dataset with actual training time and dataset sizes. Results are reported in Energy (E, meV/atom) and Force (F, meV/Å) MAE. Approximate training GPU hours are measured on 80G NVIDIA A100 GPUs.

| Dataset Name (Size) | Training Time | PubChem (33884) | | Monomers (889) | | Dimers (13896) | | Dipeptides (1025) | | SolvatedAminoAcids (52) | | Water (84) | | Qmugs (144) | | All | |
|---|---|---|---|---|---|---|---|---|---|---|---|---|---|---|---|---|---|
| | (gpu hours) | E | F | E | F | E | F | E | F | E | F | E | F | E | F | E | F |
| MACE Small | 168 | 1.41 | 35.68 | 1.04 | 17.63 | 0.98 | 16.31 | 0.84 | 25.07 | 1.60 | 38.56 | 1.67 | 28.53 | 1.03 | 41.45 | 1.27 | 29.76 |
| MACE Medium | 240 | 0.91 | 20.57 | 0.63 | 9.36 | 0.58 | 9.02 | 0.52 | 14.27 | 1.21 | 23.26 | 0.76 | 15.27 | 0.69 | 23.58 | 0.80 | 17.03 |
| MACE Large | 336 | 0.88 | 14.75 | 0.59 | **6.58** | 0.54 | 6.62 | **0.42** | 10.19 | **0.98** | 19.43 | **0.83** | 13.57 | 0.45 | 16.93 | 0.77 | 12.26 |
| E2Former 33M | 70 | **0.67** | **8.9** | **0.49** | 7.1 | **0.43** | **4.01** | 0.51 | **5.63** | 1.1 | **19.2** | 0.96 | **13.52** | **0.65** | **10.2** | **0.60** | **7.46** |

E2Former achieved state-of-the-art performance in most subsets, particularly in datasets with ample data, such as PubChem and DEShaw370-Dimers. Furthermore, compared to the MACE-Large model, E2Former achieves approximately a *fivefold* increase in training speed, thereby further validating its efficiency.

## 4.3 Meomory and Efficiency Scaling

To further probe efficiency, we evaluated computational performance in an even more extreme case: system scale. We benchmarked E2Former against MACE-Large and EquiformerV2 on systems containing up to 6,400 atoms (Table 4). E2Former consistently achieved the highest throughput, with its performance advantage becoming clearer as system size increased. At 3,200 atoms, E2Former processes data nearly three times faster than MACE-Large. Crucially, E2Former was the only model capable of handling simulations at the 6,400-atom scale, a size at which both MACE-Large and EquiformerV2 failed.

Table 4: Comparison of memory cost and efficiency on large-scale systems with up to 6,400 atoms.

| Atom Number | Memory Cost (GB) | | | Training Efficiency (Samples/Second) | | |
|---|---|---|---|---|---|---|
| | Equiformer V2-22M | MACE-Large-33M | E2Former-33M | Equiformer V2-22M | MACE-Large-33M | E2Former-33M |
| 200 | 14.2 | 6 | 3.9 | 1.81 | 5.1 | 4.5 |
| 400 | 26.7 | 10.7 | 7.6 | 1.27 | 2.91 | 4.2 |
| 800 | 53 | 23 | 16 | 0.68 | 1.61 | 2.36 |
| 1600 | - | 38 | 20 | 0.83 | 0.81 | 2.46 |
| 3200 | - | 74 | 40 | - | 0.41 | 1.2 |
| 6400 | - | - | 80 | - | - | 0.59 |

# 5 Molecular Dynamics Simulation

In this section, we demonstrate the practical utility of E2Former in molecular dynamics simulations. While machine learning methods are extensively used to predict molecular and material properties, accurately simulating the behavior of systems over extended time periods remains a significant challenge. This task requires not only precise predictions at each time step but also long-term stability and performance comparable to established methods such as DFT and empirical potential models.

We began by pretraining E2Former on a large in-house dataset derived from DeShaw (2M) [15] and GEMS [60] (2.7M), constituting a *foundational model* on machine-learning force field.

## 5.1 Small-scale Amino Acid Systems

To evaluate the model, we first performed an NVT ($T = 300$ K) simulation of an amino acid wrapped by water molecules (The structure is shown in Fig. 4(a)), totaling 253 atoms. This evaluation was conducted on a system equipped with a single NVIDIA A100 GPU

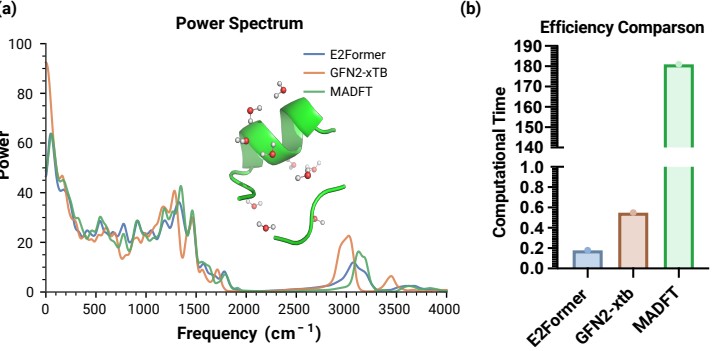

Figure 3: (a) Power spectra comparison across computational methods: E2Former (blue), GFN2-xTB (orange), and MADFT (green). The graph corresponds to a simulation at NVT ensemble, temperature $T = 300$ K, with a time step of 1 fs. A structural overlay of the simulated system is displayed for context. (b) Efficiency comparison showing computational time for E2Former, GFN2-xtb, and MADFT. E2Former demonstrates the lowest computational time. The y-axis denotes the computation time for a single frame.

and an AMD EPYC 7V13 24-core CPU. Over the course of 10,000 simulation time steps (1fs per step), we compared the trajectory 's power spectrum obtained from E2Former, CUDA-accelerated DFT [34]- namely, the MADFT software and state-of-the-art empirical potential methods GFN2-xTB [3]. Fig. 4(a) illustrates the results. The evaluation demonstrates that E2Former exhibits long-term stability in molecular simulations. The power spectrum shows that E2Former predictions align closely with those of the DFT baseline. In contrast, empirical method shows significant devia-

tions in high-frequency regions, particularly near 3,000 and 3,500 frequencies. These high-frequency components are critical as they provide insights into bond vibrations [14] and molecular stability [12], underscoring the ability of E2Former to effectively extrapolate across the molecular potential energy surface. To assess computational efficiency, we compare runtime across methods, which is depicted in Fig. 4(b). E2Former achieves a computational speed approximately 1,000 times faster than DFT, and around 2 times faster than GFN2-xTB.

## 5.2 Large-Scale 6000-atom Water Cluster

To evaluate accuracy and efficiency at larger scales, we tested our model on a 6,000-atom water cluster by analyzing atomic vibration patterns. E2Former closely reproduced the reference DFT results, accurately capturing key spectral features: low-frequency modes (below 1000 cm$^{-1}$), the H–O–H bending mode (1650 cm$^{-1}$), and O–H stretching vibrations. In contrast, MACE-Large exhibited larger deviations, particularly for high-frequency stretching modes. We further validated our approach on a more complex system—a Chignolin peptide solvated in water (approximately 2,000 atoms, Fig 6)—successfully optimizing its structure while maintaining high force prediction accuracy (0.484 kcal/mol/Å error).

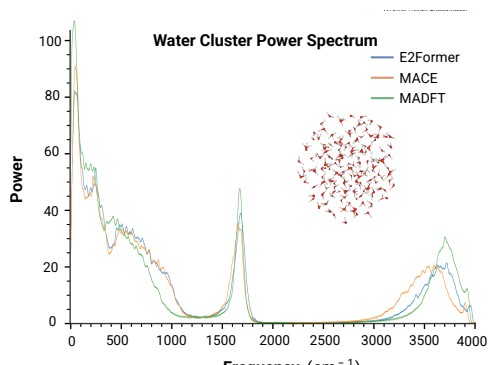

Figure 4: Power spectra comparison across computational methods: E2Former (blue), MACE (orange), and MADFT (green).

# 6 Related Work

**Invariant GNNs.** Invariant geometric GNNs have driven state-of-the-art performance in predicting molecular and crystalline properties [53, 50, 11, 26, 43, 25, 62, 48] and have been instrumental in advancing protein structure prediction [35].

**Cartesian Equivariant GNNs.** Building on invariance, Cartesian equivariant GNNs explicitly model transformations in $\mathbb{R}^3$, offering greater flexibility. These models have shown strong empirical results in similar domains [33, 51, 17, 54, 2] and have recently evolved to include Cartesian equivariant transformer layers [22].

**Spherical Equivariant GNNs.** Complementing Cartesian approaches, spherical equivariant GNNs leverage spherical tensors to naturally handle rotational symmetries, relying on the representation theory of SO(3). Recent advancements include SO(3)- and SE(3)-equivariant transformer layers [23, 41], efficient interatomic potential calculations [7, 6, 46], and optimizations that reduce convolutions in SO(3) to SO(2) [47]. These improvements have enabled strong performance in diverse applications, including geometry, physics, and chemistry [57], dynamic molecular modeling [1], and fluid mechanical modeling [58].

# 7 Conclusion and Future Work

We introduced E2Former, an efficient and scalable Transformer architecture for molecular modeling. By leveraging the Wigner $6j$ convolution, E2Former shifts computation from edges to nodes, reducing complexity from $O(|\mathcal{E}|)$ to $O(|\mathcal{V}|)$ while maintaining rotational equivariance and expressive power. E2Former demonstrated competitive performance across OC20, OC22, and SPICE benchmarks with significantly improved computational efficiency. Its scalability makes it suited for large-scale applications in biology, drug discovery, and materials science.

Future work could focus on optimizing E2Former for hardware accelerators and integrating kernelized Euclidean attention [21]. Combining Wigner-$6j$ Conv with SO(2) convolution could further bolster the model's efficiency. Scaling to the real-world all-atom protein systems will also be investigated.

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

# Appendix Table of Contents

# A Glossary of Notations

Table 5: Glossary of Notations

| Symbol | Description |
|---|---|
| $\mathcal{V}, \mathcal{E}$ | Set of nodes (vertices) $\mathcal{V}$ and edges $\mathcal{E}$ in a molecular graph. |
| $|\mathcal{V}|, |\mathcal{E}|$ | Number of nodes and edges in the graph, respectively. |
| $N$ | Often used to denote $|\mathcal{V}|$, the number of atoms (nodes). |
| $d$ | Hidden feature dimension (number of channels) at each spherical degree. |
| $s$ | Spherical dimension: the number of irreps (from $\ell = 0$ to $\ell = L$). |
| $L, L_{\max}$ | Maximum angular momentum (highest spherical degree). |
| $\ell \geq 0, m \in \{-\ell, \dots, \ell\}$ | Angular momentum quantum numbers for spherical harmonics. |
| $\mathcal{R}_m^{(\ell)}(\mathbf{r})$ | Real spherical harmonic of degree $\ell$ and order $m$, evaluated at position $\mathbf{r}$. |
| $U^{(\ell)}$ | An irreducible representation (irrep) of SO(3) at angular momentum $\ell$. |
| $\mathbf{h}_i \in \mathbb{R}^{s \times d}$ | Feature representation of node $i$, containing scalar/vector components up to spherical degree $L$. |
| $\mathbf{r}_{ij} \in \mathbb{R}^3$ | Relative position vector from node $i$ to node $j$. |
| $\otimes$ | Clebsch–Gordan (CG) tensor product of two irreps. |
| $\otimes^{6j}$ | Wigner $6j$-based tensor product (recoupling) of irreps, reorganizing the CG couplings. |
| $\alpha_{ij}$ | Attention coefficient between node $i$ and $j$ in an equivariant attention layer. |
| $C_{\ell_1 m_1, \ell_2 m_2}^{(\ell_3 m_3)}$ | Clebsch–Gordan coefficient coupling two irreps $\ell_1, \ell_2$ to an output irreps $\ell_3$. |
| $\begin{pmatrix} \ell_1 & \ell_2 & \ell_3 \\ m_1 & m_2 & m_3 \end{pmatrix}$ | Wigner $3j$ symbol (equivalently related to Clebsch–Gordan coefficients). |
| $\begin{Bmatrix} j_1 & j_2 & j_3 \\ j_4 & j_5 & j_6 \end{Bmatrix}$ | Wigner $6j$ symbol, governing recoupling of three angular momenta in different orders. |
| $D^{(\ell)}(R)$ | Wigner $D$-matrix describing how spherical harmonics of degree $\ell$ transform under rotation $R$. |
| SO(3) | 3D rotation group; E2Former is equivariant to transformations in SO(3). |
| E2Former | The proposed **E**fficient and **E**quivariant Trans**former** architecture. |
| Wigner $6j$ Conv | The core convolution module leveraging Wigner $6j$ recoupling to shift edge-based operations to nodes. |

# B Attention-based Wigner $6j$ Convolution

We describe the algorithmic procedure for constructing the Attention-based Wigner $6j$ Convolution introduced in the main text. Assume that attention coefficients $\alpha$ have been precomputed, either through `Query-Key` inner products or via MLP-based attention mechanisms such as those in Equiformer v2 [42]. We begin by precomputing the spherical harmonics $\mathcal{R}^{(\ell)}$ up to a maximum degree $L$ based on the input positions $\mathbf{r}$.

For each degree $k = 0, \ldots, L$, we compute an attention-weighted tensor product between the input features $\mathbf{h}$ and the corresponding spherical harmonics $\mathcal{R}^{(k)}$. This intermediate representation is then modulated by the attention weights $\alpha$. The resulting tensors are subsequently recoupled using a Wigner $6j$ tensor product, where each recoupling path is parameterized by the Wigner $6j$ coefficients—offering a more flexible alternative to the standard Clebsch–Gordan coupling.

Finally, summing over all degrees $k$ yields the output irreducible representations (irreps), which are used to update the node embeddings.

---

**Algorithm 1** Wigner $6j$-Based Attention

---
1: **Input:** Positions $\mathbf{r} \in \mathbb{R}^{N \times 3}$, input features $\mathbf{h} \in \mathbb{R}^{N \times H \times d \times s}$, attention weights $\alpha \in \mathbb{R}^{N \times N \times H}$ ($H$ for number for heads, $s$ for the number for spherical dimension, $d$ for the number of hidden channels), maximum order $L$
2: **Output:** Output features $\mathbf{h}_{\text{out}} \in \mathbb{R}^{N \times H \times d \times s}$
3: **Step 1: Precompute Spherical Harmonics**
4: **for** $\ell = 0$ **to** $L$ **do**
5:    $\mathcal{R}^{(\ell)} \leftarrow \texttt{SH}(\ell, \mathbf{r})$
6: **end for**
7: **Step 2: Compute pairwise distances**
8: $D_{ij} \leftarrow \|\mathbf{r}_i - \mathbf{r}_j\|_2$                           *(pairwise Euclidean distance)*
9: **Step 3: Compute Wigner $6j$ tp**
10: **for** $k = 0$ **to** $L$ **do**
11:    Compute intermediate tensor product:

$$\mathbf{T}_k \leftarrow \texttt{clebsch\_gorden\_tp}(\mathbf{h}, \mathcal{R}^{(k)})$$

12:    $\alpha_{ijh}^{(k)} \leftarrow \alpha_{ijh}/(D_{ij})^k$                     *(entrywise power normalization)*
13:    Apply attention weights:

$$\mathbf{T}_k \leftarrow \texttt{einsum}(\text{“ijh, jhds} \rightarrow \text{ihds"}, \alpha^{(k)}, \mathbf{T}_k)$$

14:    Recouple terms using Wigner $6j$ symbols:

$$\mathbf{C}_k \leftarrow \texttt{Wigner6jTP}(\mathbf{T}_k, \mathcal{R}^{(L-k)})$$

15:    Update output: $\mathbf{h}_{\text{out}} \leftarrow \mathbf{h}_{\text{out}} + (-1)^k \binom{L}{k} \mathbf{C}_k$
16: **end for**

---

*Remark* B.1. The upshot is that *(i)* the node-local spherical harmonics $\mathcal{R}^{(u)}(\vec{r}_i)$ can be precomputed once for each node $i$, and *(ii)* the partial sum $\sum_j \alpha_{ij}(\mathbf{h}_j \otimes \mathcal{R}^{(\ell-u)}(\vec{r}_j))$ can be treated as a single node-based operation. Thus, the number of tensor products is controlled by $|\mathcal{V}|$ (the number of nodes) rather than $|\mathcal{E}|$ (the number of edges). This is precisely the core reason E2Former achieves improved scalability compared to conventional SO(3)-equivariant Transformers that do edge-level spherical harmonic products.

# C Wigner $3j$ and $6j$ Symbols

The Wigner $3j$ and $6j$ symbols are fundamental constructs in the representation theory of the Lie group SU(2), intrinsically linked to the theory of angular momentum in quantum mechanics. These symbols emerge as crucial transformation coefficients when decomposing tensor products of irreducible representations of SU(2). They precisely encode the symmetry properties inherent in such decompositions, thereby providing a powerful computational framework for problems involving coupled representations.

## C.1 The Wigner $3j$ Symbol: Definition and Core Properties

The Wigner $3j$ symbol is denoted by

$$\begin{pmatrix} j_1 & j_2 & j_3 \\ m_1 & m_2 & m_3 \end{pmatrix}, \tag{C.1}$$

where $j_i$ are representation labels (angular momenta) and $m_i$ are their respective components (magnetic quantum numbers). For the symbol to be non-zero, two primary selection rules must be satisfied:

1. Conservation of the $m$ quantum number: $m_1 + m_2 + m_3 = 0$.

2. Triangle inequalities for the $j$ quantum numbers: $|j_1 - j_2| \leq j_3 \leq j_1 + j_2$, and its cyclic permutations. This ensures that the three angular momenta can form a closed vector triangle.

The explicit algebraic form of the $3j$ symbol (due to Racah) is given by:

$$\begin{pmatrix} j_1 & j_2 & j_3 \\ m_1 & m_2 & m_3 \end{pmatrix} = \delta_{m_1+m_2+m_3,0}(-1)^{j_1-j_2-m_3}$$

$$\times \sqrt{\frac{(j_1 + j_2 - j_3)!(j_1 - j_2 + j_3)!(-j_1 + j_2 + j_3)!}{(j_1 + j_2 + j_3 + 1)!}}$$

$$\times \sqrt{\prod_{k=1}^{3}(j_k + m_k)!(j_k - m_k)!}$$

$$\times \sum_z \frac{(-1)^z}{z!(j_1 + j_2 - j_3 - z)!(j_1 - m_1 - z)!(j_2 + m_2 - z)!}$$

$$\times \frac{1}{(j_3 - j_2 + m_1 + z)!(j_3 - j_1 - m_2 + z)!}, \tag{C.2}$$

where the summation over the integer $z$ is constrained such that all factorial arguments remain non-negative. The initial $\delta$ factor enforces the $m$-conservation rule.

The $3j$ symbols possess several important symmetry properties:

- Even permutation of columns leaves the symbol unchanged: $\begin{pmatrix} j_1 & j_2 & j_3 \\ m_1 & m_2 & m_3 \end{pmatrix} =$ $\begin{pmatrix} j_2 & j_3 & j_1 \\ m_2 & m_3 & m_1 \end{pmatrix} = \ldots$

- Odd permutation of columns introduces a phase factor $(-1)^{j_1+j_2+j_3}$: $\begin{pmatrix} j_2 & j_1 & j_3 \\ m_2 & m_1 & m_3 \end{pmatrix} =$ $(-1)^{j_1+j_2+j_3}\begin{pmatrix} j_1 & j_2 & j_3 \\ m_1 & m_2 & m_3 \end{pmatrix}.$

- Time-reversal symmetry (negation of all $m_i$ values):

$$\begin{pmatrix} j_1 & j_2 & j_3 \\ -m_1 & -m_2 & -m_3 \end{pmatrix} = (-1)^{j_1+j_2+j_3}\begin{pmatrix} j_1 & j_2 & j_3 \\ m_1 & m_2 & m_3 \end{pmatrix}. \tag{C.3}$$

Furthermore, they satisfy crucial orthogonality relations, fundamental for their role as transformation coefficients:

$$\sum_{m_1,m_2} (2j_3 + 1)\begin{pmatrix} j_1 & j_2 & j_3 \\ m_1 & m_2 & m_3 \end{pmatrix}\begin{pmatrix} j_1 & j_2 & j_3' \\ m_1 & m_2 & m_3' \end{pmatrix} = \delta_{j_3,j_3'}\delta_{m_3,m_3'}, \tag{C.4}$$

where the sum is over all allowed $m_1, m_2$ for fixed $j_1, j_2$. Another form is:

$$\sum_{j_3,m_3} (2j_3 + 1)\begin{pmatrix} j_1 & j_2 & j_3 \\ m_1 & m_2 & m_3 \end{pmatrix}\begin{pmatrix} j_1 & j_2 & j_3 \\ m_1' & m_2' & m_3 \end{pmatrix} = \delta_{m_1,m_1'}\delta_{m_2,m_2'}. \tag{C.5}$$

## C.2 The Wigner $6j$ Symbol: Recoupling Coefficients

The Wigner $6j$ symbol, denoted $\{\dots\}$, addresses the recoupling of three angular momenta. It arises when transforming between different sequential coupling schemes for combining three irreducible representations of $\mathrm{SU}(2)$. For instance, coupling $j_1$ and $j_2$ to an intermediate $j_{12}$, then coupling $j_{12}$ with $j_3$ to a final $J$, versus coupling $j_2$ and $j_3$ to $j_{23}$, then $j_1$ with $j_{23}$ to $J$.

It is defined through a sum over products of four $3j$ symbols:

$$\begin{Bmatrix} j_1 & j_2 & j_3 \\ j_4 & j_5 & j_6 \end{Bmatrix} = \sum_{m_1,\dots,m_6} \sum_{m_1',\dots,m_3'} (-1)^{\sum_{k=1}^{6}(j_k - m_k)} \begin{pmatrix} j_1 & j_2 & j_3 \\ m_1 & m_2 & m_3 \end{pmatrix} \begin{pmatrix} j_1 & j_5 & j_6 \\ m_1' & -m_5 & m_6 \end{pmatrix}$$

$$\times \begin{pmatrix} j_4 & j_2 & j_6 \\ m_4 & m_2' & -m_6' \end{pmatrix} \begin{pmatrix} j_4 & j_5 & j_3 \\ -m_4' & m_5' & m_3' \end{pmatrix}, \tag{C.6}$$

where the $m$ and $m'$ indices are appropriately summed while respecting the $3j$ symbol selection rules

The $6j$ symbol is directly related to the Racah $W$-coefficient by a phase factor:

$$\begin{Bmatrix} j_1 & j_2 & j_3 \\ j_4 & j_5 & j_6 \end{Bmatrix} = (-1)^{j_1+j_2+j_4+j_5} W(j_1 j_2 j_5 j_4; j_3 j_6). \tag{C.7}$$

The Racah $W$-coefficient, $W(abcd; ef)$, is the transformation coefficient between schemes $((a,b)e,d)c$ and $(a,(b,d)f)c$. Thus, the $6j$ symbol effectively captures the algebraic structure of associativity in tensor products of representations.

Symmetries of the $6j$ symbol are extensive:

- Invariance under any permutation of its columns.
- Invariance under the exchange of upper and lower arguments in any two columns, e.g.:
$$\begin{Bmatrix} j_1 & j_2 & j_3 \\ j_4 & j_5 & j_6 \end{Bmatrix} = \begin{Bmatrix} j_4 & j_5 & j_3 \\ j_1 & j_2 & j_6 \end{Bmatrix}.$$

These 24 symmetries reflect the tetrahedral symmetry associated with the symbol, often visualized using Yutsis graphs. Each set $\{j_1, j_2, j_3\}$, $\{j_1, j_5, j_6\}$, $\{j_4, j_2, j_6\}$, and $\{j_4, j_5, j_3\}$ must satisfy the triangle inequalities for the $6j$ symbol to be non-zero.

An important orthogonality relation (one of several, including the Biedenharn-Elliott identity) is:

$$\sum_{j_3} (2j_3 + 1)(2j_6 + 1) \begin{Bmatrix} j_1 & j_2 & j_3 \\ j_4 & j_5 & j_6 \end{Bmatrix} \begin{Bmatrix} j_1 & j_2 & j_3 \\ j_4 & j_5 & j_6' \end{Bmatrix} = \delta_{j_6, j_6'} \Delta(j_1, j_5, j_6) \Delta(j_4, j_2, j_6). \tag{C.8}$$

Here, $\Delta(a,b,c) = 1$ if $a, b, c$ satisfy the triangle inequalities, and $0$ otherwise.

A remarkable connection to geometry is provided by the Ponzano–Regge asymptotic formula. For large $j$ values, it relates the $6j$ symbol to the geometry of a tetrahedron whose edge lengths are $j_k + \frac{1}{2}$:

$$\begin{Bmatrix} j_1 & j_2 & j_3 \\ j_4 & j_5 & j_6 \end{Bmatrix} \approx \frac{1}{\sqrt{12\pi V}} \cos\left( \sum_{k=1}^{6} (j_k + \tfrac{1}{2})\theta_k + \frac{\pi}{4} \right), \tag{C.9}$$

where $V$ is the volume of the tetrahedron and $\theta_k$ are the dihedral angles. This formula bridges quantum angular momentum algebra with semi-classical geometric concepts.

# D Main Theorem Proofs

## D.1 Binominal Local Expansion

**Claim D.1** (Multiplicity-One of $V^{(L_{\max})}$ [61, 8]). *The irreducible representation $V^{(L_{\max})}$ appears with multiplicity one in the decomposition of $V^{(l_1)} \otimes \cdots \otimes V^{(l_k)}$. That is,*

$$V^{(l_1)} \otimes \cdots \otimes V^{(l_k)} \cong \bigoplus_L N_L V^{(L)}, \quad \text{with } N_{L_{\max}} = 1.$$

**Claim D.2** (Schur's Lemma [24]). *Let $V, W$ be irreducible representations of a group $G$ over an algebraically closed field. If $T : V \to W$ is a $G$-equivariant linear map, then either $T = 0$, or $V \cong W$ and $T$ is a scalar multiple of the identity.*

**Claim D.3** (Recoupling via Wigner $6j$ Symbols [49]). *For any two CG coupling trees $\tau$ and $\tau'$, the associated contraction operations are related by a unitary transformation:*

$$\mathcal{C}_{\tau'} = R_{\tau' \leftarrow \tau} \mathcal{C}_\tau,$$

*where $R_{\tau' \leftarrow \tau}$ is a product of Wigner $6j$ matrices acting on intermediate coupling channels.*

**Lemma D.4** (Ordering-Invariance of Maximally Coupled Representation). *Let $\mathbf{h}_1^{(l_1)}, \dots, \mathbf{h}_k^{(l_k)} \in V^{(l_1)} \otimes \cdots \otimes V^{(l_k)}$ denote a sequence of irreducible features, where each $V^{(l_i)}$ is the irreducible representation of $\mathrm{SO}(3)$. Let $L_{\max} := \sum_{i=1}^k l_i$. Then for any binary CG coupling tree $\tau$, the projection of the coupled tensor onto the $L_{\max}$ subspace,*

$$\left[ \mathcal{C}_\tau \left( \mathbf{h}_1^{(l_1)}, \dots, \mathbf{h}_k^{(l_k)} \right) \right]^{(L_{\max})},$$

*is invariant to the choice of $\tau$.*

*Proof.* The tensor product $V^{(l_1)} \otimes \cdots \otimes V^{(l_k)}$ admits a decomposition into irreducible components of the form $\bigoplus_L N_L V^{(L)}$, where $N_L$ denotes the multiplicity of the spin-$L$ representation. It is a classical fact (Claim D.1) that the maximal total spin $L_{\max} := \sum_i l_i$ appears with multiplicity one. Consequently, the subspace $V^{(L_{\max})}$ is uniquely defined up to a basis, and any projection onto it is one-dimensional in each magnetic subspace.

Each binary CG coupling tree $\tau$ defines a recursive contraction $\mathcal{C}_\tau$ via successive applications of the Clebsch–Gordan tensor product. For any two such trees $\tau$ and $\tau'$, the corresponding coupled features are related by a transformation $\mathcal{C}_{\tau'} = R_{\tau' \leftarrow \tau} \mathcal{C}_\tau$, where $R_{\tau' \leftarrow \tau}$ is constructed from a sequence of Wigner $6j$ symbols. This recoupling matrix is unitary (by Claim D.3) and acts on the space of intermediate angular momentum.

Since $V^{(L_{\max})}$ appears with multiplicity one, Schur's lemma ( Claim D.2) implies that any $\mathrm{SO}(3)$-equivariant transformation, such as $R_{\tau' \leftarrow \tau}$, must act as a scalar on this subspace:

$$R_{\tau' \leftarrow \tau}\big|_{V^{(L_{\max})}} = \lambda_{\tau', \tau} \cdot \mathrm{Id}, \quad \text{with } |\lambda_{\tau', \tau}| = 1.$$

By adopting a fixed phase convention (e.g., the Condon–Shortley convention for CG coefficients), we can choose $\lambda_{\tau', \tau} = 1$, yielding

$$\left[ \mathcal{C}_\tau (\mathbf{h}_1^{(l_1)}, \dots, \mathbf{h}_k^{(l_k)}) \right]^{(L_{\max})} = \left[ \mathcal{C}_{\tau'} (\mathbf{h}_1^{(l_1)}, \dots, \mathbf{h}_k^{(l_k)}) \right]^{(L_{\max})}.$$

An equivalent interpretation is that $V^{(L_{\max})}$ can be realized as the totally symmetric, trace-free subspace of the rank-$L_{\max}$ tensor formed from the inputs $\mathbf{h}_1^{(l_1)}, \dots, \mathbf{h}_k^{(l_k)}$. Since full symmetrization commutes with all permutations and parenthesizations, the final projected tensor is independent of the coupling order. This concludes the proof. $\square$

**Theorem D.5** (Projection identity for spherical harmonics). *Let $\mathbf{r}_{ij} := \mathbf{r}_i - \mathbf{r}_j \in \mathbb{R}^3 \setminus \{\vec{0}\}$ and let $\mathcal{R}^{(l)}(\vec{r})$ denote the degree-$l$ spherical harmonic. For every integer $\ell \geq 0$,*

$$\mathcal{R}^{(\ell)}(\mathbf{r}_{ij}) = \left[ \left( \mathcal{R}^{(1)}(\mathbf{r}_{ij}) \right)^{\otimes \ell} \right]^{(\ell)},$$

*where $(\cdot)^{\otimes \ell}$ is the $\ell$-fold tensor product and the $[ \cdot ]^{(\ell)}$ denotes the orthogonal projector onto the irreducible $\mathrm{SO}(3)$ subspace of total angular momentum $\ell$.*

*Proof.* The function $\mathcal{R}^{(1)}(\mathbf{r}_{ij})$ transforms according to the irreducible (vector) representation $D^{(1)}$ of $SO(3)$. Hence the tensor power satisfies

$$\left(\mathcal{R}^{(1)}(\mathbf{r}_{ij})\right)^{\otimes \ell} \in \bigotimes{}^{\ell} D^{(1)} \cong \bigoplus_{l=0}^{\ell} D^{(l)} \otimes \mathbb{C}^{m_l},$$

where $m_l$ is the multiplicity of $D^{(l)}$ in the Clebsch–Gordan decomposition. Applying the projector $[\,\cdot\,]^{(\ell)}$ extracts the $D^{(\ell)}$ summand, yielding a rank-$\ell$ tensor that transforms in the same irrep as $\mathcal{R}^{(\ell)}$.

Because $D^{(\ell)}$ is irreducible, Schur's lemma (Claim D.2) implies that any two non-zero intertwiners from $D^{(\ell)}$ to itself differ by a scalar. Thus the projected tensor must be proportional to the degree-$\ell$ spherical harmonic:

$$\left[\left(\mathcal{R}^{(1)}(\mathbf{r}_{ij})\right)^{\otimes \ell}\right]^{(\ell)} = c_\ell\, \mathcal{R}^{(\ell)}(\mathbf{r}_{ij}).$$

To determine the constant $c_\ell$, evaluate both sides on the north-pole direction $\vec{z} = (0,0,1)$. In the Condon–Shortley convention $\mathcal{R}_{\pm 1}^{(1)}(\vec{z}) = 0$ and $\mathcal{R}_0^{(1)}(\vec{z}) = \sqrt{3/(4\pi)}$, so the only non-vanishing component of the tensor power corresponds to the highest-weight vector $|\ell, \ell\rangle$. A direct Clebsch–Gordan calculation (or induction on $\ell$) shows that its norm matches that of $\mathcal{R}_\ell^{(\ell)}(\hat{z}) = \sqrt{(2\ell + 1)/(4\pi)}$, fixing $c_\ell = 1$. Since both sides transform identically under rotations, equality for one direction implies equality for all directions, completing the proof. $\square$

**Theorem D.6** (Binomial Local Expansion). *Let $\vec{r}_i, \vec{r}_j \in \mathbb{R}^3$ denote the positions of two nodes $i, j$, and let $\vec{r}_{ij} = \vec{r}_j - \vec{r}_i$. For each integer $\ell \geq 1$, let $\mathcal{R}^{(\ell)}(\cdot)$ denote the real-valued spherical harmonics of order $\ell$. Then,*

$$\mathcal{R}^{(\ell)}(\vec{r}_{ij}) = \left[\left(\mathcal{R}^{(1)}(\vec{r}_j) - \mathcal{R}^{(1)}(\vec{r}_i)\right)^{\otimes \ell}\right]^{(\ell)} = \sum_{u=0}^{\ell} (-1)^{\ell - u} \binom{\ell}{u} \left[\mathcal{R}^{(u)}(\vec{r}_j) \otimes \mathcal{R}^{(\ell - u)}(\vec{r}_i)\right]^{(\ell)},$$

*where $[\cdot]^{(\ell)}$ denotes projection onto the irreducible subspace of total angular momentum $\ell$ via Clebsch–Gordan decomposition. Equality holds up to a normalization constant depending on the basis choice for $\mathcal{R}^{(\ell)}$ and the CG convention.*

*Proof.* By Theorem D.5, the spherical harmonic $\mathcal{R}^{(\ell)}(\mathbf{r}_{ij})$ can be constructed by projecting the $\ell$-fold tensor product of first-order harmonics onto the irreducible subspace of total angular momentum $\ell$:

$$\mathcal{R}^{(\ell)}(\mathbf{r}_{ij}) = \left[\left(\mathcal{R}^{(1)}(\mathbf{r}_j) - \mathcal{R}^{(1)}(\mathbf{r}_i)\right)^{\otimes \ell}\right]^{(\ell)}.$$

This expression follows from the identity $\mathbf{r}_{ij} = \mathbf{r}_j - \mathbf{r}_i$ and the fact that $\mathcal{R}^{(1)}$ is linear in spatial coordinates.

Expanding the tensor power via the multinomial binomial rule yields

$$\left(\mathcal{R}^{(1)}(\mathbf{r}_j) - \mathcal{R}^{(1)}(\mathbf{r}_i)\right)^{\otimes \ell} = \sum_{u=0}^{\ell} (-1)^{\ell - u} \binom{\ell}{u} \sum_{P \in \Pi_u} T_P,$$

where each term $T_P$ corresponds to a specific ordering $P \in \Pi_u$ of the tensor product, containing exactly $u$ factors of $\mathcal{R}^{(1)}(\mathbf{r}_j)$ and $\ell - u$ factors of $\mathcal{R}^{(1)}(\mathbf{r}_i)$. The total number of such orderings is $|\Pi_u| = \binom{\ell}{u}$.

Applying the linear projection operator $[\cdot]^{(\ell)}$ to both sides gives:

$$\left[\left(\mathcal{R}^{(1)}(\mathbf{r}_j) - \mathcal{R}^{(1)}(\mathbf{r}_i)\right)^{\otimes \ell}\right]^{(\ell)} = \sum_{u=0}^{\ell} (-1)^{\ell - u} \binom{\ell}{u} \sum_{P \in \Pi_u} [T_P]^{(\ell)}.$$

At this point, we invoke the ordering-invariance lemma (Lemma D.4), which states that the projection of a tensor product onto the highest angular momentum subspace (here, $\ell$) is invariant under permutation of the tensor factors. Therefore, the projected tensors $[T_P]^{(\ell)}$ are identical for all $P \in \Pi_u$, and depend only on the multiplicities of $\mathcal{R}^{(1)}(\mathbf{r}_j)$ and $\mathcal{R}^{(1)}(\mathbf{r}_i)$ within the product.

We may thus replace the sum over $P \in \Pi_u$ with a multiplicity factor times a single representative term. Letting $T_{\mathrm{rep}} := (\mathcal{R}^{(1)}(\mathbf{r}_j))^{\otimes u} \otimes (\mathcal{R}^{(1)}(\mathbf{r}_i))^{\otimes(\ell-u)}$, we obtain:

$$\sum_{P \in \Pi_u} [T_P]^{(\ell)} = \binom{\ell}{u} [T_{\mathrm{rep}}]^{(\ell)} = \binom{\ell}{u} \left[ (\mathcal{R}^{(1)}(\mathbf{r}_j))^{\otimes u} \otimes (\mathcal{R}^{(1)}(\mathbf{r}_i))^{\otimes(\ell-u)} \right]^{(\ell)}.$$

Finally, note that $(\mathcal{R}^{(1)}(\mathbf{r}))^{\otimes u}$ contains irreducible components of order up to $u$, and the highest such component is $\mathcal{R}^{(u)}(\mathbf{r})$. Projecting the expression onto total angular momentum $\ell$ thus yields:

$$\mathcal{R}^{(\ell)}(\mathbf{r}_{ij}) = \sum_{u=0}^{\ell} (-1)^{\ell-u} \binom{\ell}{u} \left[ \mathcal{R}^{(u)}(\mathbf{r}_j) \otimes \mathcal{R}^{(\ell-u)}(\mathbf{r}_i) \right]^{(\ell)},$$

which concludes the derivation. $\qquad\square$

## D.2 Wigner $6j$ Recoupling and Node-Based Factorization

Here, we formalize the proof of the Theorem 3.3 for completeness.

**Setup:** We consider a typical $SO(3)$-equivariant Transformer layer that performs message passing from each node $j$ in the neighborhood of $i$ using the tensor product $\mathbf{h}_j \otimes \mathcal{R}^{(\ell)}(\vec{r}_{ij})$. Symbolically,

$$\mathbf{h}_i^{\text{new}} = \sum_{j \in \mathcal{N}(i)} \alpha_{ij}\big(\mathbf{h}_j \otimes \mathcal{R}^{(\ell)}(\vec{r}_{ij})\big).$$

**Goal:** To show that the expensive edge-based computation over $(i, j)$ can be reorganized so that the *tensor product* portion (or at least the dominating part of it) depends only on node $i$ *and* a separate node $j$ portion. This is achieved by:

$$\mathcal{R}^{(\ell)}(\vec{r}_{ij}) \mapsto \sum_{u=0}^{\ell} \Big[\mathcal{R}^{(u)}(\vec{r}_i)\Big] \otimes^{6j} \Big[\mathcal{R}^{(\ell-u)}(\vec{r}_j)\Big],$$

together with a rearrangement (via Wigner $6j$) of $\mathbf{h}_j$ inside the product.

**Theorem D.7** (Node-Based Factorization via Wigner $6j$). *Let $\mathbf{h}_j \in \mathbb{R}^d$ be the feature of node $j$, and let $\alpha_{ij}$ be any scalar weight (e.g. an attention coefficient). In the $SO(3)$-equivariant layer:*

$$\sum_{j \in \mathcal{N}(i)} \big(\mathbf{h}_j \otimes \mathcal{R}^{(\ell)}(\vec{r}_{ij})\big),$$

*we can reorganize $\mathcal{R}^{(\ell)}(\vec{r}_{ij})$ into node-i and node-j parts by Theorem D.6 and then apply Wigner $6j$ recoupling to obtain:*

$$\sum_{j \in \mathcal{N}(i)} \big(\mathbf{h}_j \otimes \mathcal{R}^{(\ell)}(\vec{r}_{ij})\big) = \sum_{u=0}^{\ell}(-1)^{\ell-u}\binom{\ell}{u}\Big[\mathcal{R}^{(u)}(\vec{r}_i)\Big] \otimes^{6j} \Big(\sum_{j \in \mathcal{N}(i)} \big[\mathbf{h}_j \otimes \mathcal{R}^{(\ell-u)}(\vec{r}_j)\big]\Big).$$

*Proof.* We commence with the definition of the $SO(3)$-equivariant node convolution for node $i$. Substituting the Binomial Local Expansion for $\mathcal{R}^{(\ell)}(\mathbf{r}_{ij})$ yields:

$$\mathbf{h}_i = \sum_{j \in \mathcal{N}(i)} \mathbf{h}_j \otimes \left(\sum_{u=0}^{\ell}(-1)^{\ell-u}\binom{\ell}{u}\Big[\mathcal{R}^{(u)}(\mathbf{r}_i) \otimes \mathcal{R}^{(\ell-u)}(\mathbf{r}_j)\Big]^{(\ell)}\right). \tag{D.1}$$

Invoking the linearity of the Clebsch-Gordan tensor product $\otimes$ with respect to its second argument, and subsequently interchanging the order of the finite summations (over $j \in \mathcal{N}(i)$ and $u \in [0, \ell]$), Eq. (D.1) is rewritten as:

$$\mathbf{h}_i = \sum_{u=0}^{\ell}(-1)^{\ell-u}\binom{\ell}{u}\sum_{j \in \mathcal{N}(i)}\left(\mathbf{h}_j \otimes \Big[\mathcal{R}^{(u)}(\mathbf{r}_i) \otimes \mathcal{R}^{(\ell-u)}(\mathbf{r}_j)\Big]^{(\ell)}\right). \tag{D.2}$$

We now focus on the term within the summation over $j$:

$$T_{j,u} = \mathbf{h}_j \otimes \Big[\mathcal{R}^{(u)}(\mathbf{r}_i) \otimes \mathcal{R}^{(\ell-u)}(\mathbf{r}_j)\Big]^{(\ell)}. \tag{D.3}$$

Let $U_A = \mathbf{h}_j$, $U_C = \mathcal{R}^{(u)}(\mathbf{r}_i)$, and $U_B = \mathcal{R}^{(\ell-u)}(\mathbf{r}_j)$ represent the respective irreducible representations. The term $T_{j,u}$ signifies a specific coupling scheme: $U_C$ (irrep $u$) and $U_B$ (irrep $\ell - u$) are first coupled, and their product is projected onto the irreducible component transforming as irrep $\ell$, denoted $[U_C \otimes U_B]^{(\ell)}$. Subsequently, $U_A$ is coupled with this resulting tensor of irrep $\ell$. This scheme corresponds to $(U_A \otimes (U_C \otimes U_B)^{L_{CB}=\ell})$, where $L_{CB} = \ell$ is the fixed intermediate angular momentum.

The theory of angular momentum recoupling, governed by Wigner $6j$ symbols (as per Definition 2.3 ), allows for the reordering of tensor product operations while preserving the final irreducible content. We seek to transform $T_{j,u}$ into a form where $U_A$ and $U_B$ are coupled first. Specifically, we apply a

recoupling to achieve the order $U_C \otimes^{6j} (U_A \otimes U_B)$. The transformation from $(U_A \otimes (U_C \otimes U_B)^{L_{CB}=\ell})$ to $(U_C \otimes (U_A \otimes U_B)^{L_{AB}})$ (for any resulting total angular momentum) is a standard result in Wigner-Racah calculus. The operator $\otimes^{6j}$ denotes that the CG tensor product is performed with path weights modified by the appropriate Wigner $6j$ symbol, which accounts for this change in coupling pathway. Note that the intermediate coupling of $\mathcal{R}^{(u)}(\mathbf{r}_i)$ and $\mathcal{R}^{(\ell-u)}(\mathbf{r}_j)$ results in a constrained irrep $\ell$ (governed by the projection operator); this gives rise to a constrained Wigner $6j$ coupling, where the constraint (the intermediate angular momentum being $\ell$) is inherently managed by the $6j$ coefficients encapsulated within the $\otimes^{6j}$ operation. Thus, we can write:

$$\mathbf{h}_j \otimes \left[ \mathcal{R}^{(u)}(\mathbf{r}_i) \otimes \mathcal{R}^{(\ell-u)}(\mathbf{r}_j) \right]^{(\ell)} = \mathcal{R}^{(u)}(\mathbf{r}_i) \otimes^{6j} \left( \mathbf{h}_j \otimes \mathcal{R}^{(\ell-u)}(\mathbf{r}_j) \right). \tag{D.4}$$

Substituting Eq. (D.4) into Eq. (D.2):

$$\mathbf{h}_i = \sum_{u=0}^{\ell} (-1)^{\ell-u} \binom{\ell}{u} \sum_{j \in \mathcal{N}(i)} \left( \mathcal{R}^{(u)}(\mathbf{r}_i) \otimes^{6j} \left( \mathbf{h}_j \otimes \mathcal{R}^{(\ell-u)}(\mathbf{r}_j) \right) \right). \tag{D.5}$$

The term $\mathcal{R}^{(u)}(\mathbf{r}_i)$ is independent of the summation index $j$. The operation $\otimes^{6j}$, like the standard tensor product $\otimes$, is linear in its second argument with respect to summation. Thus, $\mathcal{R}^{(u)}(\mathbf{r}_i) \otimes^{6j}$ can be factored out of the sum over $j$:

$$\mathbf{h}_i = \sum_{u=0}^{\ell} (-1)^{\ell-u} \binom{\ell}{u} \left( \mathcal{R}^{(u)}(\mathbf{r}_i) \otimes^{6j} \left( \sum_{j \in \mathcal{N}(i)} \mathbf{h}_j \otimes \mathcal{R}^{(\ell-u)}(\mathbf{r}_j) \right) \right). \tag{D.6}$$

This expression is the factorized form of the SO(3)-equivariant node convolution as stated in the theorem. □

# E   Proof of Equivariance

We now formally establish that the Wigner 6j convolution is equivariant under the action of the rotation group $\mathrm{SO}(3)$. Define the convolutional output as follows:

$$f(\mathbf{x}) = \sum_{u=0}^{\ell} \left( \mathcal{R}^{(u)}(\mathbf{r}_i) \otimes^{6j} \left( \alpha_{ij} \left( \mathbf{h}_j \otimes \mathcal{R}^{(\ell-u)}(\mathbf{r}_j) \right) \right) \right).$$

We analyze the transformation of this expression under the rotation $R \in \mathrm{SO}(3)$. The transformed output is:

$$f(R \cdot \mathbf{x}) = \sum_{u=0}^{\ell} \left( D^{(u)}(R) \mathcal{R}^{(u)}(\mathbf{r}_i) \otimes^{6j} \left( \alpha_{ij} \left( D^{(a)}(R) \mathbf{h}_j \otimes D^{(\ell-u)}(R) \mathcal{R}^{(\ell-u)}(\mathbf{r}_j) \right) \right) \right).$$

Let us define the intermediate quantity:

$$S_u \triangleq \sum_{j \in \mathcal{N}(i)} \alpha_{ij} \, \mathbf{h}_j \otimes \mathcal{R}^{(\ell-u)}(\mathbf{r}_j).$$

Under rotation, this transforms as:

$$\sum_{j \in \mathcal{N}(i)} \alpha_{ij} \left( D^{(a)}(R) \mathbf{h}_j \otimes D^{(\ell-u)}(R) \mathcal{R}^{(\ell-u)}(\mathbf{r}_j) \right) = \left( D^{(a)}(R) \otimes D^{(\ell-u)}(R) \right) S_u.$$

Substituting into the convolution expression yields:

$$f(R \cdot \mathbf{x}) = \sum_{u=0}^{\ell} \left( D^{(u)}(R) \mathcal{R}^{(u)}(\mathbf{r}_i) \otimes^{6j} \left( \left( D^{(a)}(R) \otimes D^{(\ell-u)}(R) \right) S_u \right) \right).$$

Since the Wigner 6j recoupling tensor $\otimes^{6j}$ is $\mathrm{SO}(3)$-equivariant (as the Wigner coefficients are invariant under rotation), we may commute the group action:

$$D^{(u)}(R) \mathcal{R}^{(u)}(\mathbf{r}_i) \otimes^{6j} \left( \left( D^{(a)}(R) \otimes D^{(\ell-u)}(R) \right) S_u \right) = D_{\mathrm{out}}(R) \left( \mathcal{R}^{(u)}(\mathbf{r}_i) \otimes^{6j} S_u \right),$$

where $D_{\mathrm{out}}(R)$ denotes the output representation under $\mathrm{SO}(3)$. Therefore:

$$f(R \cdot \mathbf{x}) = \sum_{u=0}^{\ell} D_{\mathrm{out}}(R) \left( \mathcal{R}^{(u)}(\mathbf{r}_i) \otimes^{6j} S_u \right) = D_{\mathrm{out}}(R) f(\mathbf{x}).$$

This completes the proof of $\mathrm{SO}(3)$-equivariance.

# F Time Complexity

We analyze the computational complexity of the Wigner-$6j$ convolution, with particular attention to its dependence on the number of nodes $\mathcal{V}$, the number of channels $C$, and the maximum angular momentum $L$. The dominant cost stems from angular momentum coupling via tensor products and Wigner-$6j$ recoupling, while the linear self-mix step introduces an additional quadratic dependence on $C$. This analysis aligns with the derivations in Appendix C of the eSCN paper for $SO(3)$ convolution.

Let $\mathcal{V}$ denote the total number of nodes (e.g., atoms), and $C$ the number of channels per irreducible representation. The algorithm involves iterating over all valid angular momentum triplets $(L_1, L_2, L_o)$ up to order $L$, subject to triangle inequality constraints.

> **(1) Tensor Product:** Each tensor product between irreps $(L_1, L_2)$ incurs a cost
> $$\Theta(C \cdot L_1 \cdot L_2)$$
>
> **(2) Wigner-$6j$ and CG Multiplication:** The cost of recoupling the tensor via Wigner-$6j$ coefficients is
> $$\Theta(C \cdot L_1 \cdot L_2 \cdot L_o)$$
>
> **(3) Summation Step:** The final projection onto $L_o$ includes a summation step costing
> $$\Theta(C \cdot L_o)$$
>
> per valid triplet.

The summation step is implemented as follows:

```
outputs = self._sum_tensors([out for ins, out in zip(instructions, outputs)
                            if ins.i_out == i_out])
```

The **per-node** complexity of Wigner-$6j$ convolution is therefore:

$$\sum_{L_1=0}^{L} \sum_{L_2=0}^{L} \sum_{L_o=0}^{L} (CL_1L_2 + CL_1L_2L_o + CL_o) = \Theta(CL^6)$$

**Linear Self-Mix.** After the convolution, a channel-mixing layer (e.g., an MLP) acts independently on each irrep. This operation mixes $C$ channels across each irrep of order $L_o$, with complexity:

$$\sum_{L_o=0}^{L} \Theta(C^2 L_o) = \Theta(C^2 L^2)$$

**Total Complexity.** Summing over all nodes $\mathcal{V}$, the total computational complexity is:

$$\boxed{\Theta(|\mathcal{V}|CL^6 + |\mathcal{V}|C^2L^2)}$$

# G  Additional Lemmas

**Lemma G.1** (Many–Body Reduction). *Using Wigner $6j$ convolution, a multi-atomic cluster expansion can be evaluated in $O(|V|)$ time instead of $O(|\mathcal{E}|)$.*

*Proof.* We decompose the MACE architecture into two computational stages and show that each admits nodewise computation at $O(|V|)$ cost.

**Stage 1: $SO(3)$-Equivariant Convolution.**  The convolutional component of MACE computes, for each node $i \in V$,

$$\mathbf{A}_i := \sum_{j \in \mathcal{N}(i)} \mathbf{h}_j \otimes \mathcal{R}^{(\ell)}(\mathbf{r}_{ij}),$$

which has naive complexity $O(|\mathcal{E}|)$ over all nodes due to edge enumeration.

**Stage 2: Self Tensor Product and Projection.**  The self tensor product $(\mathbf{A}_i)^{\otimes p}$ and projection $[\cdot]^{(L)}$ are purely local operations, independent of neighbors. Since they apply once per node, they require $O(|V|)$ total time.

**Key Step: Nodewise Refactorization via Wigner $6j$.**  Using Theorem 3.3, we invoke the following identity:

$$\sum_{j \in \mathcal{N}(i)} \mathbf{h}_j \otimes \mathcal{R}^{(\ell)}(\mathbf{r}_{ij}) = \sum_{u=0}^{\ell} \mathcal{R}^{(u)}(\mathbf{r}_i) \otimes^{6j} \left( \sum_{j \in \mathcal{N}(i)} \mathbf{h}_j \otimes \mathcal{R}^{(\ell-u)}(\mathbf{r}_j) \right),$$

where $\otimes^{6j}$ denotes a tensor contraction via Wigner $6j$ recoupling.

Define global moment tensors $\mathbf{M}^{(\ell-u)} := \sum_{j \in V} \mathbf{h}_j \otimes \mathcal{R}^{(\ell-u)}(\mathbf{r}_j)$. These are precomputable in $O(|V|)$ time, and reused across all $i \in V$. The refactorized form becomes

$$\psi_i = \sum_{u=0}^{\ell} \mathcal{R}^{(u)}(\mathbf{r}_i) \otimes^{6j} \mathbf{M}^{(\ell-u)},$$

which is independent of the neighborhood structure and thus computable per node in constant time (for fixed $\ell$).

**Complexity.**  The precomputation of $\{\mathbf{M}^{(\ell-u)}\}_{u=0}^{\ell}$ costs $O(|V|)$. The nodewise contractions via $\otimes^{6j}$ are constant-time per node for bounded $\ell$, yielding an overall convolutional cost of $O(|V|)$. Since the subsequent self tensor product and projection are already node-local, the total complexity of MACE becomes $O(|V|)$.

$\square$

# H   Technical Details Behind E2Former

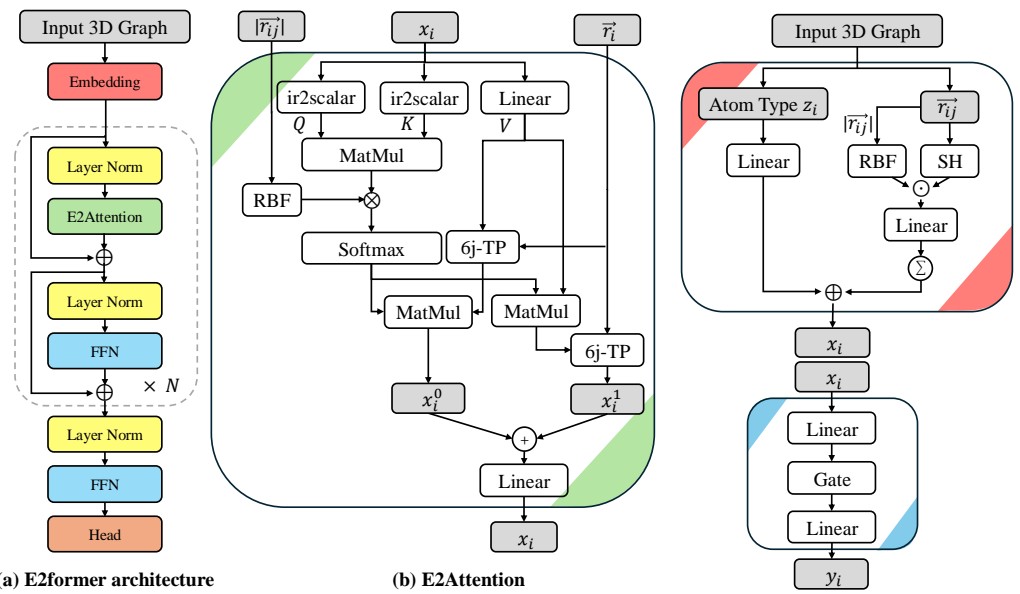

**(a) E2former architecture**    **(b) E2Attention**

Figure 5: Overview of the E2Former architecture. (a) The main network alternates E2Attention blocks with feedforward layers, repeatedly refining node embeddings from a 3D molecular graph. (b) Within each E2Attention block, scalarized queries/keys (via `ir2scalar`) are combined with distance-dependent features (`RBF`) and convolutions (`6j-TP`), updating the node embeddings equivariantly. (c) The final readout incorporates atomic types and radial/spherical expansions (`RBF`, `SH`) into a gated projection that produces the per-atom output $y_i$.

In our development, we not only want mathematical integrity but also practical usefulness. Thus, E2Former is not solely characterized by Wigner-$6j$ conv, but rather as an architecture that integrates both efficiency and significant engineering components. The benefit of Wigner-$6j$ conv enables us to reallocate the computational budget to increase expressivity elsewhere, such as adding more layers, wider hidden dimensions, or more attention heads.

**Architecture Overview.**    The E2FORMER architecture adheres to the general transformer paradigm, commencing with an embedding layer that generates initial E(3)-equivariant node features. These features are subsequently refined through a stack of E2former blocks, denoted as `TransBlock` in our implementation. Each `TransBlock` employs a pre-normalization strategy, structured as: Normalization → Transformer Layer → Residual Connection → Normalization → Equivariant Feed-Forward Network (FFN) → Residual Connection.

**Initial Embedding and Feature Representation.**    The generation of initial equivariant node features, $\mathbf{h}_i^{(0)}$, for each node $i$ (e.g., an atom) with input coordinates and type, is performed by an *Initial Equivariant Embedding* module. This module constructs features by aggregating information from the local neighborhood. Interatomic distances are encoded using radial basis functions (RBFs), while relative positions, $\mathbf{r}_j - \mathbf{r}_i$, are represented using spherical harmonics, $\mathcal{R}_m^{(l)}(\vec{p}_j - \vec{p}_i)$, up to a specified maximum degree $l_{\max}$. The resultant features, $\mathbf{h}_i^{(0)}$, comprise a collection of irreducible representations (irreps) of SO(3). Within each `TransBlock`, *Equivariant Normalization* layers are applied before both the attention and FFN sub-layers. These layers operate by normalizing features independently within each irrep channel, which is crucial for stabilizing training and enhancing model performance.

**Equivariant Feed-Forward Networks.**    Subsequent to the attention mechanism, node features are processed by an *Equivariant Feed-Forward Network*. E2FORMER accommodates a variety of FFN

types, configurable via the `ffn_type` parameter. These include standard equivariant Multi-Layer Perceptrons (MLPs) that operate on spherical harmonic coefficients (e.g., `FeedForwardNetwork_s2`, `FeedForwardNetwork_s3`, potentially incorporating grid-based non-linearities inspired by eSCN and EquiformerV2 [47, 42]), as well as explicit many-body interaction modules [6] that integrate equivariant two-body or three-body tensor products. This modular design permits tailored feature processing contingent upon the specific demands of the task.

**E2Attention Mechanism.** The central component enabling feature interaction in E2FORMER is the *E2Attention* mechanism, which builds upon the Wigner-$6j$ Convolution to update node embeddings. A key implementation consideration lies in the treatment of positional information. While many spherical EGNNs normalize relative positions, this practice may discard essential directional cues in Wigner-$6j$ Convolution, which is inherently node-centric. To preserve relative geometric information, we retain unnormalized absolute positions in the convolution, and instead apply normalization during the attention coefficient computation.

**All-Order Attention Paths.** E2Attention explicitly models and adaptively aggregates contributions from multiple *spherical harmonic orders*. Under the `attn_type="all-order"` configuration, the mechanism includes a zero-order (scalar) path that captures isotropic interactions, a first-order (vector) path using Wigner-$6j$ Convolution configured for order-1 interactions, and higher-order paths (e.g., second-order for $l = 2$) to model more complex anisotropic effects.

**Computation of Attention Weights.** Attention weights $\alpha_{ij}$ are computed from projected scalar queries and keys derived from $\mathbf{h}_i$ and $\mathbf{h}_j$, enriched by radial basis function (RBF) embeddings of $\|\mathbf{r}_{ij}\|$ and optionally, learnable embeddings of atomic types $z_i, z_j$. Geometric information can be incorporated into attention in various ways, such as through the `tp_type="dot_alpha"` configuration, which directly integrates spherical harmonics into the attention score computation.

**Gated Aggregation of Orders.** A critical component of E2Attention is the *Gated Aggregation of Orders*. The contributions from the zero-order ($\mathbf{m}_{ij}^{(0)}$), first-order ($\mathbf{m}_{ij}^{(1)}$), and second-order ($\mathbf{m}_{ij}^{(2)}$) pathways are adaptively combined via a learnable gating mechanism. Specifically, the scalar ($l = 0$) components of the central node's features $\mathbf{h}_i$ are processed through a small MLP to produce gating coefficients $g_i^{(0)}, g_i^{(1)}, g_i^{(2)}$. The aggregated message from neighbor $j$ to node $i$ is given by

$$\mathbf{m}_{ij} = g_i^{(0)} \odot \mathbf{m}_{ij}^{(0)} + g_i^{(1)} \odot \mathbf{m}_{ij}^{(1)} + g_i^{(2)} \odot \mathbf{m}_{ij}^{(2)}.$$

The updated node feature $\mathbf{h}_i'$ is then computed by summing the attention-weighted messages from all neighbors:

$$\mathbf{h}_i' = \sum_j \alpha_{ij} \mathbf{m}_{ij}.$$

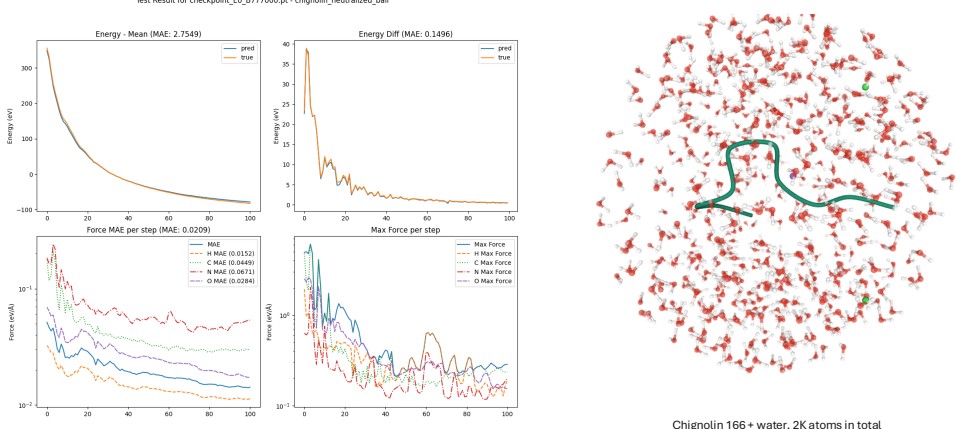

Figure 6: Force MAE on a system of around 2000 atoms, where a chignolin peptide is wrapped with water.

# I  Addtional Experiments

## I.1  QM9 Results

We additionally evaluated our method on the QM9 dataset as a quality check. Due to computational constraints, we report results on three representative energy metrics: $U_0$, HOMO, and LUMO. E2Former demonstrates competitive performance compared to Equiformer V2 and its predecessor Equiformer, while GotenNet [2] achieves the best overall results. Nonetheless, we emphasize that our model is primarily designed for larger systems, whereas QM9 represents a relatively small-scale benchmark.

Table 6: Performance on QM9 ($U_0$, HOMO, LUMO)

| Method | $U_0$ | HOMO | LUMO |
|---|---|---|---|
| E2Former | 6.43 | 14.2 | 13.6 |
| GotenNet | 3.37 | 13.4 | 12.2 |
| Equiformer V2 | 6.17 | 14.4 | 13.3 |
| Equiformer | 6.59 | 15.4 | 14.7 |

## I.2  Comparison with Equiformer V2

To ensure a better comparison with the Equiformer V2 model, we aligned the experimental settings between E2Former and Equiformer V2. Specifically, we adopted identical hyperparameters and architecture configurations for both models, including the number of layers, maximum angular momentum orders, and radial basis functions. This alignment guarantees a controlled comparison and eliminates confounding factors arising from differing model capacities or training settings.

Table 7 summarizes the performance in terms of energy mean absolute error (E), force mean absolute error (F), and inference speed (measured in samples per second). The results indicate that E2Former not only achieves better accuracy on both energy and force predictions but also delivers significantly faster inference speed under matched conditions.

Table 7: Comparison between E2Former and Equiformer V2 under identical hyperparameter settings.

| Method | E | F | Number of Layers | $L_{\max}$ | $M_{\max}$ | Inference Speed (samples/sec) |
|---|---|---|---|---|---|---|
| E2Former | 20.5 | 270 | 12 | 3 | 2 | 34 |
| Equiformer V2 | 23.47 | 296 | 12 | 3 | 2 | 22 |

# J  Hyperparameters

Table 8: Hyperparameter Configuration for E2Former on OC20, OC22, and SPICE

| Hyperparameter | E2Former 33M | E2Former 67M | Description |
|---|---|---|---|
| *— General Training Settings —* | | | |
| `optim.lr_initial` | 0.00015 | 0.0002 | Initial learning rate for the optimizer. |
| `optim.batch_size` | 128 for OC20/OC22 and 48 for SPICE | 64 | Training batch size. |
| *— Model Architecture —* | | | |
| `model.backbone.encoder_embed_dim` | 256 | 256 | Embedding dimension for each node. |
| `model.backbone.hidden_size` | 256 | 256 | Hidden size for intermediate layers. |
| `model.backbone.num_layers` | 6 | 12 | Number of E2Former layers. |
| `model.backbone.max_neighbors` | 20 | 20 | Max neighbors per node for message passing. |
| `model.backbone.irreps_node_embedding` | 256x0e+256x1e+256x2e+256x3e | 256x0e+256x1e+256x2e+256x3e | Irreps for node embeddings up to $\ell = 3$. |
| `model.backbone.irreps_head` | 16x0e+16x1e+16x2e+16x3e | 16x0e+16x1e+16x2e+16x3e | Irreps for the final head up to $\ell = 3$. |
| `model.backbone.attn_scalar_head` | 16 | 16 | Size of scalar attention head projections. |
| `model.backbone.num_attn_heads` | 32 | 32 | Number of multi-head attentions per layer. |
| `model.backbone.number_of_basis` | 256 | 256 | Number of radial basis functions. |
| `model.backbone.max_radius` | 12 for OC20/22 and 5 for SPICE | 12 for OC20/22 and 5 for SPICE | Cutoff radius for local neighborhood. |
| `model.backbone.alpha_drop` | 0.05 | 0.05 | Drop rate for alpha (e.g., attention dropout). |
| `model.backbone.drop_path_rate` | 0.05 | 0.05 | Stochastic depth/drop path rate. |
| `model.backbone.basis_type` | `gaussiansmear` | `gaussiansmear` | Type of radial embedding (Gaussian smearing). |
| `model.backbone.norm_layer` | `layer_norm_sh` | `layer_norm_sh` | Normalization layer type (LayerNorm in spherical basis). |
| `model.backbone.attn_type` | `all-order` | `all-order` | Attention mechanism covering all spherical orders. |
| `model.backbone.tp_type` | `dot_alpha` | `dot_alpha` | Type of tensor product (dot + learned scale). |
| `model.backbone.ffn_type` | `s2` | `s2` | Type of feed-forward network in each block. |

