# OpenReview forum: "E2Former: An Efficient and Equivariant Transformer with Linear-Scaling Tensor Products"
_NeurIPS.cc/2025/Conference — NeurIPS 2025 spotlight_

### Official Review · Reviewer_7r4S · 2025-06-03

**Clarity:** 2
**Significance:** 4
**Originality:** 4
**Rating:** 6
**Confidence:** 5

**Summary:**

This paper presents the Wigner 6j Convolution (Wigner 6j Conv), a spherical equivariant neural network architecture that leverages Wigner 6-j symbols to shift the computation of tensor products from edges to nodes, enhancing the computational efficiency.

The authors’ response has fully addressed all of my concerns, resulting in a logical and coherent manuscript. Given the high efficiency and broad application prospects of the proposed method (Wigner6j-conv), I recommend a **Strong Accept**. However, I encourage the authors to carefully revise the manuscript to ensure that the narrative and exposition are fully commensurate with the quality of this excellent work.

**Questions:**

See weakness.

**Ethical Concerns:**

["NO or VERY MINOR ethics concerns only"]

**Final Justification:**

The authors’ response has fully addressed all of my concerns, resulting in a logical and coherent manuscript. Given the high efficiency and broad application prospects of the proposed method (Wigner6j-conv), I recommend a **Strong Accept**. However, I encourage the authors to carefully revise the manuscript to ensure that the narrative and exposition are fully commensurate with the quality of this excellent work.

**Limitations:**

Yes.

**Paper Formatting Concerns:**

The spacing between the caption and the body of the table seems to have been adjusted, especially Tab. 3, which lacks the required blank line. This needs to be corrected if the article is accepted.

**Quality:**

4

**Strengths And Weaknesses:**

### S. 1 Interesting Motivation
Losslessly reducing the number of tensor product calculations is a valuable research discovery.

### W.1 Ambiguous Presentation
The presentation and mathematical writing of this paper have a lot of room for improvement, which has forced me to spend a lot of effort trying to clarify it. Here is my understanding. Please clarify it:
1. In Def. 2.1, why  normalization factor $k_m^{(l)}$ does not have subscript $m$ when $m\leq 0$?
2. In Def 2.2, why does the summarization traverse superscript $l_1, l_2$?
3. In Line 122, should it be $Y^{(1)}(\vec r_{ij})\propto Y^{(1)}(\vec r_{ic})-Y^{(1)}(\vec r_{jc})$? There is a lack of decentralization and it is not strict equality.
4. Fig. 1(a) in this paper is fancy and difficult to understand, and needs to be modified. The orange dashed line is difficult to distinguish, why not use different colors to distinguish. The background color of the formula is not distinguishable, the font is inconsistent, and text "xx dependent" is not conducive to understanding.

The above are some small details that greatly affect the reading experience. If I have misunderstood them and caused errors in the evaluation, please forgive me. Next, I need to point out that there are big problems with the overall presentation logic of this article.
1. The core of this article is to reduce the number of tensor product calculations, which is somewhat different from reducing complexity. It is recommended to adjust the description.
2. Simple mathematics takes up a lot of space, including most of Sec. 2, and the proofs of Thms. 3.2 and 3.3, which are trivial and can be placed directly in the appendix. In particular, the authors introduce spherical harmonics in trigonometric form, which is completely unnecessary - for the binomial expansion, spherical harmonics in the form of $x,y,z$ are more helpful.
3. As an article whose title emphasizes the new model "E2former", but the main body of the article hardly discusses the model - the highlight module in Appendix H should be moved forward to the main text (exchanged with the mathematical proof). In addition, the introduction of the model in Appendix H lacks a clear mathematical form, which is inappropriate and should be clearly given.

Frankly speaking, it is very abrupt to give the proof of Thms. 2.2-2.3 in the main text, and the completion of these two paragraphs is out of tune with the completion of other paragraphs. The most important model design part is given in the appendix, making the argument and the statement unable to support each other - it is like an article recycled many times and lost its original coherence and character.

### W.2 Missing Relevant Reference
I suggest moving "Related Work" after "Introduction". In addition, it does not reflect the relevance to this article. I think the following changes should be included.
1. Combine all existing introductions to invariant and equivariant models. Focus on the pros and cons of tensor products, such as strong expressiveness but time-consuming. In addition, the latest spherical-scalarization models, such as SO3KRATES [NC'24], HEGNN[NeurIPS'24] and GotenNet[ICLR'25], should be discussed. They completely abandon tensor products, which is much more efficient than the one in this paper. What are the benefits of retaining tensor products in this paper?
2. The design of equivariant transformers should be discussed separately, such as SE3-Transformer [NeurIPS'20], SO3KRATES [NeurIPS'22], MEAN [ICLR'23], EquiformerV2 [NeurIPS'23], Geoformer [NeurIPS'23], GotenNet [ICLR'25], DPA2 [NPJ'24], eSEN [arXiv:2502.12147].

### W3. Experimental Design and Results
1. In the results of OC-22 in the main text, E2former does not have an advantage, which cannot illustrate the potential of this method. In addition, without seeing the ablation experiment, it is difficult to determine which module is effective.
2. The results in Tab. 5 are inferior to GotenNet, and lack comparisons with other indicators, which makes people worry about the effect of the model on small-scale systems.
3. Although the authors say that "our model is primarily designed for larger systems", the three experiments in the main text are not large enough to be called large-scale. It is recommended to refer to VINE-GATr [ICLR'25 MLMP], Geometric Hyena [ICML'25] for additional experiments.

---

> ### Author Rebuttal · Authors · 2025-07-28
>
> Thank you for your detailed review and helpful feedback.   We address your concerns below and describe the improvements we will make in our revision.
>
>
> **W1.1** Thanks for spotting those notation errors. You are right—we should use $k_m^{(l)}$ consistently throughout. We will fix these typos immediately. Also, following Reviewer ejRh’s suggestion, we will update our terminology to ‘solid spherical harmonics’ to reflect the unnormalized form we use.
>
> **W1.2** Our original writing attempts to convey that $m$ is tied to $l$. To avoid confusion, we will modify the summation to be  $\sum_{m_1 = - l_1}^{l_1} \sum_{m_2 = - l_2}^{l_2}$.
>
> **W1.3**   $Y^{(1)}(\vec{r}_{ij}) = Y^{(1)}(\vec{r}_i) - Y^{(1)}(\vec{r}_j)$ holds exactly for solid spherical harmonics, as clarified with Reviewer ejRh. Here, $Y^{(1)}(\vec{r}_i)$ follows the standard convention of denoting positions relative to a predefined origin.
>
> **W1.4 Figure 1:** We appreciate your feedback on visual clarity. We have carefully redrawn the figure with distinct colors for different components, improved font consistency, and replaced the "xx dependent" labels with more descriptive terminology. We apologize that we cannot provide the revised figure here due to NeurIPS restrictions on sharing updated PDFs or external links, but these improvements will be incorporated in the final version of our paper.
>
> **W1.5 Complexity Clarification:** Thank you for pointing out the difference between reducing tensor product calculations and overall complexity. We apologize for not highlighting this more clearly and will revise our wording to be more precise.
>
> **W1.6 Overall Presentation Logic:** We sincerely thank the reviewer for the detailed and constructive feedback. We appreciate the reviewer's suggestions on the organization and agree that enhancing the clarity and coherence of our manuscript is important.
>
> We initially chose to emphasize the mathematical derivations in the main text, given that the Wigner 6j convolution is a central contribution enabling the efficiency of our model, as demonstrated by the 7x-30x speedup in Section 4.1. However, we fully acknowledge the reviewer’s concern that a clearer connection between theory and practical architecture would greatly improve readability. Therefore, we will incorporate an **overview and diagram of the E2Former architecture** into the main body, highlighting key design choices like gated aggregation, and will clearly point readers to additional details provided in the appendix.
>
> Regarding the proofs of Theorems 3.2 and 3.3, we respectfully believe they represent meaningful contributions. The introduction of the Wigner 6j convolution specifically addresses the reduction of tensor product calculations from $O(|\mathcal{E}|)$ to $O(|\mathcal{V}|)$.  The theoretical justifications for these proofs are crucial: without the proof of Theorem 3.2, the factorization would only work for spherical harmonics of order 1, and it would not be clear why the binomial theorem applies in irreps space. Similarly, without introducing Wigner 6j symbols as done in the proof of Theorem 3.3, the decomposition from edge-level to node-level TPs would not be possible (for example $\sum_j h_j \otimes Y(r_i) \otimes Y(r_j)$ would still require edge-scaling TPs).  The derivation of these results requires careful treatment of angular momentum coupling theory, which we believe will interest the broader ML community. However, we fully acknowledge the reviewer’s point that the proofs may appear abrupt in the current presentation. To address this, we will add an introductory explanation at the beginning of Section 3 to provide clear context and intuition. Additionally, we will greatly **simplify the proof sketches** to improve readability and ensure smoother integration within the main text.
>
> We greatly appreciate the reviewer’s suggestions and believe these modifications will enhance the accessibility of our manuscript.
>
>
> **W2.1 Section placement:** We agree with this suggestion and will move the “Related Work” section immediately after the introduction to better contextualize our contributions.
>
> **W2.2 Missing comparisons and discussion:** We appreciate your insightful feedback, and to address your valuable suggestions, we will incorporate an expanded and detailed discussion of spherical-scalarization methods.
>
> To frame our perspective within the broader context of EGNN, we note that scalarization is a special case of the tensor product: its characteristic steerable vector‑to‑scalar interaction is naturally expressed as a TP. For example, when an odd‑type vector interacts with a scalar, the operation can be written as $1\times L_o \otimes 1\times 0_e \rightarrow 1\times L_o$ (with similar derivations applicable to other operations). Studying the full tensor‑product setting is therefore still meaningful. Our aim is not to claim that tensor‑product models are superior, but to **optimize** a well‑established framework—one that underlies many state‑of‑the‑art molecular‑modeling systems (e.g., OMat24; ECD, ICLR ’25; QH9, NeurIPS ’23). Moreover, tensor‑product methods remain the preferred approach for systems with periodic boundary conditions (OMat24).
>
>
> **W2.3 Equivariant transformer discussion:** We appreciate this suggestion and will create a dedicated subsection discussing equivariant transformer architectures to provide a more comprehensive overview of the field.
>
>
> **W3** We appreciate the reviewer’s feedback on our experimental design. Our primary goal is to demonstrate that E2Former achieves performance comparable to state-of-the-art spherical equivariant methods (such as Equiformer V2), but with significantly enhanced efficiency. The key contribution of our work is this scalability, enabling large-scale simulations previously impractical for most spherical equivariant models.
>
>
> **W3.1: OC-22 Performance and Ablation Study** While E2Former does not set a new state-of-the-art on the OC-22 benchmark, it achieves competitive results with substantially greater efficiency. E2Former’s training requires only 1,500 GPU hours, a threefold reduction compared to the 4,500 hours needed for the leading Equiformer V2 model. This highlights our method’s core design goal: enabling scalability and efficiency rather than extracting marginal performance gains on existing benchmarks.
>
> As requested, we conducted an ablation study on the OC20 2M dataset. The table below compares our complete model to several variants, demonstrating that each architectural component contributes to either improved accuracy or enhanced efficiency.
>
>
> |  | Energy MAE | Force MAE | Inference Speed |
> | --- | --- | --- | --- |
> | w/o Gated Aggregation | 270 | 21.2 | 42 |
> |  w/o S2 Activation | 284 | 22.4 | 62 |
> | w/o wigner 6j conv | 273 | 21.8 | 20 |
> | Full | 275 | 21.9 | 62 |
>
> **W3.2: Small-Scale System Performance**  We appreciate the reviewer's observation on small-scale benchmarks. Our model is specifically optimized for efficiency on large-scale systems, where its primary advantages are most evident. The QM9 benchmark was included in the appendix as a quality check rather than a primary evaluation metric. As expected, E2Former's performance on this smaller benchmark is comparable to similar architectures like Equiformer and EquiformerV2, which aligns with our design objectives. We thank the reviewer for raising this point and will work to further enhance performance on small-scale benchmarks in the future.
>
> **W3.3: Real-World Large-Scale Experiments**
> Following the reviewer's suggestion, we added new experiments on large-scale systems. We trained the model on a large dataset consist of diverse molecules with DFT force labels and evaluated it using MD simulations of systems containing up to 6,000 atoms. Our results closely match DFT calculations and outperform baseline methods.
>
> We first tested the efficiency and scalability on those systems. E2Former consistently achieved the highest throughput for systems up to 6,400 atoms, with its performance advantage becoming clearer as system size increased. At 3,200 atoms, E2Former processes data nearly three times faster than MACE. Importantly, E2Former is the only model capable of handling simulations at 6,400 atoms, a scale where other models fail.
>
> | Atom Number | Memory Cost (GB) |  |  | Training Efficiency (Samples/Second)↑ |  |  |
> | --- | --- | --- | --- | --- | --- | --- |
> |  | Equiformer V2-22M | MACE-33M | E2Former-33M | Equiformer V2-22M | MACE-33M | E2Former-33M |
> | 200 | 14.2 | 6 | 3.9 | 1.81 | 5.1 | 4.5 |
> | 400 | 26.7 | 10.7 | 7.6 | 1.27 | 2.91 | 4.2 |
> | 800 | 53 | 23 | 16 | 0.68 | 1.61 | 2.36 |
> | 1600 | - | 38 | 20 | 0.83 | 0.81 | 2.46 |
> | 3200 | - | 74 | 40 | - | 0.41 | 1.2 |
> | 6400 | - | - | 80 | - | - | 0.59 |
>
> To check both accuracy and speed, we tested our model on a 6,000-atom water cluster and looked at the vibration patterns of the atoms. E2Former matched the reference DFT results very closely, capturing all the important features: low-frequency movements (below 1000 cm⁻¹), the H–O–H bending motion (1650 cm⁻¹), and most importantly, the O–H stretching pattern. MACE-Large, on the other hand, had significant errors, especially for the high-frequency stretching modes. We also tested on a more complex system - a Chignolin peptide in water (~2,000 atoms) - and successfully optimized its structure while keeping force predictions highly accurate (0.484 kcal/mol/A error).
>
> *Note: We apologize that we cannot share the figure due to NeurIPS restrictions. The table below describes the key peaks from our analysis.*
> | Vibrational Mode | Method | Peak Frequency (cm−1) | Relative Error vs. DFT (%) |
> | --- | --- | --- | --- |
> | **H-O-H Bending** | CUDA DFT (Reference) | ~1650 | 0.0% |
> |  | **E2Former-Pretrain (Ours)** | **~1670** | **~1.2%** |
> |  | MACE-large-Pretrain | ~1700 | ~3.0% |
> | **O-H Stretching** | CUDA DFT (Reference) | ~3550 | 0.0% |
> |  | **E2Former-Pretrain (Ours)** | **~3550** | **~0.0%** |
> |  | MACE-large-Pretrain | ~3400 | ~4.2% |

---

> > ### Comment · Reviewer_7r4S · 2025-08-01
> >
> > I appreciate the authors’ thoughtful responses and have carefully reviewed both the reviewers’ comments and the subsequent clarifications. It appears there is a broad consensus on the significance of this work, particularly in relation to the reduction of tensor product computations via Wigner 6j. My initial concerns leading to a less positive review were primarily focused on the narrative structure of the paper rather than its technical content, and it seems that Reviewer ZFTk raised similar points.
> >
> > The paper is organized around two main contributions: first, the reduction of tensor product calculations through Wigner 6j; second, the design of E2former, an architecture that leverages this acceleration. If we conceptualize the acceptance bar for a NeurIPS submission as a score of 1, I would estimate that each component independently merits a score of around 0.8. The central challenge then becomes how to combine these two contributions such that their sum exceeds the threshold for acceptance. As written, however, the paper reads more as a juxtaposition of two related—but not fully integrated—ideas. This gives the impression that, although the work has substantial potential, its current presentation does not fully realize a coherent and compelling narrative.
> >
> > Based on my understanding, the use of Wigner 6j appears to offer a plug-and-play type of acceleration that is largely independent of the E2former architecture. E2former itself is an extension of Equiformer (as referenced in Figure 4), and it is structurally complete without necessarily requiring Wigner 6j. The ablation results provided (Section W3.1) demonstrate that omitting Wigner 6j convolution yields very similar accuracy to the original, in line with the assertion of “operational equivalence.” There is even a slight performance improvement, which is likely attributable to enhanced training stability. This is a promising observation. Therefore, it may be valuable to further explore the generalizability of the Wigner 6j approach—perhaps by incorporating it into other architectures like TFN, MACE, or Equiformer on smaller-scale benchmarks—to more robustly demonstrate its applicability.
> >
> > Regarding the introduction of E2former: as a model, it appears to be a self-contained enhancement, which prompts the question of why a new model was designed instead of extending existing architectures. Does this reflect challenges in scaling existing architectures to larger parameter regimes? It would help to clarify what specific advantages E2former offers relative to Equiformer. Further discussion on these points would improve understanding of both the practical and theoretical impact of the proposed approaches.
> >
> > In general, I encourage the authors to refine the narrative to achieve a more cohesive integration of the two key innovations presented in the paper. Additionally, providing straightforward experimental evidence—even on a small scale, such as evaluating one or two molecules from the MD17 or MD22 datasets—would help to substantiate the claims. **Should the authors address these points, I would be inclined to raise my evaluation.**
> >
> >
> > ---
> > Lastly, I would like to offer some suggestions regarding the discussion of related literature. The statement that "scalarization is a special case of the tensor product" may not be entirely appropriate. In model design, greater generality does not always translate to better performance. For instance, there are many scenarios where EGNN outperforms TFN, and GotenNet surpasses EquiformerV2, which suggests that formal generalization does not necessarily correlate with practical performance.
> >
> > If the authors are uncertain about how best to frame this discussion, I propose the following approach: briefly highlight the speed and effectiveness of “spherical-scalarization methods.” Then, drawing on the discussion presented in D-spanning [A], clarify that although these scalarization methods can be highly efficient, the tensor product approach remains theoretically complete. Emphasize that accelerating the full tensor product operation is therefore still an important and relevant challenge. This perspective more accurately situates the contributions of the current work within the broader context of equivariant model design.
> >
> > [A] Nadav Dym and Haggai Maron. On the universality of rotation equivariant point cloud networks.

---

> ### Author Response · Authors · 2025-08-04
>
> We sincerely thank the reviewer for their insightful and constructive feedback, which significantly enhances the clarity and depth of our manuscript.
>
> **1. The Wigner 6j Convolution as a General, Numerically-Faithful Speedup.** We agree with the reviewer regarding the potential general applicability of the Wigner 6j convolution as a numerically faithful solution to accelerate tensor products in various equivariant architectures.
>
> To rigorously substantiate this claim, we performed an additional targeted ablation study on numeric equivalence. Specifically, we compared outputs from a 12-layer, 384-hidden-channel E2Former initialized identically with and without the Wigner 6j convolution (untrained, single random seed). Results demonstrate numerical equivalence within single-precision limits:
>
> - **Energy MAE:** 2.13e-5 ± 4.35e-5 (relative error: 3.84e-4 ± 8.67e-4)
> - **Force MAE:**  5.90e-8 ± 22.1e-8 (relative error: 5.43e-5 ± 2.05e-4)
>
> *Note: This experiment was performed with random system of 50 atoms using a 12-layer model with a 384-channel hidden dimension, initialized with the same random seed. The model was intentionally left untrained to isolate the numerical equivalence of the operation itself.*
>
> **2. Architectural Rationale, and Plug and Play Implementation**. We agree with the reviewer’s perspective on the broader potential of our Wigner 6j convolution as a 'plug-and-play' solution. We are especially thankful for your perceptive question about our architectural choices, as it allows us to articulate a key insight that was not sufficiently highlighted before: although the mathematical operation is equivalent, the complex, edge-centric designs of existing models make a simple drop-in replacement difficult, which is what motivated the development of E2Former.
>
> In Equiformer, the architecture is inherently **edge-centric and hierarchical**. To compute an attention weight ($\alpha_{ij}$), it first combines node features ($h_i, h_j$) into an edge feature $m_{ij}$, followed by computing an expensive tensor product between $m_{ij}$ and geometric information $Y(r_{ij})$, followed by a set of linear and non-linear layers. Then Equiformer calculates a second stage tp $\alpha_{ij} m_{ij} \otimes Y(r_{ij})$, **further compounding model’s complexity**.
>
> E2Former, by contrast, is inherently **node-centric**. We use a standard, efficient inner-product attention where the weights ($\alpha_{ij}$) are calculated *without* any tensor products. The Wigner 6j tensor product is decoupled from the attention and is performed efficiently at the node level. This architectural choice is crucial; it preserves the node-scaling tensor-products of our method.
>
> We appreciate the reviewer’s suggestion regarding integration into existing architectures. While we acknowledge, as noted by Reviewer ejRh, that implementing in modern architectures like Equiformer or MACE is non-trivial, we commit to integrating Wigner 6j into the simpler architecture (SE(3)-Transformer variant/TFN) and will report results in the final revised manuscripts.

---

> ### Author Response · Authors · 2025-08-04
>
> **3. Enhanced Experimental Validation on Challenging Systems.** As requested, we provide new evaluations on the MD22 benchmark to demonstrate E2Former's strong performance on complex small-scale systems, validating our design choices. E2Former is highly competitive with specialized SOTA models.
>
> |  | ATATCGCG - (Force MAE) | ATAT - (Force MAE) |
> | --- | --- | --- |
> | Equiformer | 0.1252 | 0.0960 |
> | ViSNet | 0.1563 | 0.1070 |
> | MACE | 0.1153 | 0.0992 |
> | Allegro | 0.1280 | 0.0952 |
> | SO(3) Karates | 0.332 | 0.216 |
> | Gotennet | **0.0824** | **0.0632** |
> | E2Former (Ours) | *0.1062* | *0.0823* |
>
> *Note: Selected from MD22 as representative systems due to limited rebuttal time; full table will be in the revised manuscripts.*
>
> We also wish to clarify a minor point: we apologize for previously misunderstanding the reviewer's intended meaning regarding decentralization. Indeed, our implementation preprocesses coordinates by subtracting the center of mass, ensuring the system is centralized before training. We will clearly highlight this detail in the revised manuscript.
>
> **4. Positioning within Related Literature**. We sincerely appreciate the reviewer’s insightful suggestions and constructive feedback on framing our discussion of related literature. We fully agree that explicitly highlighting the strengths of spherical-scalarization methods and clearly positioning our work within the broader landscape of equivariant model design would significantly enhance our manuscript’s narrative clarity. Indeed, as perceptively noted by the reviewer, specialized scalarization approaches such as ViSNet and GotenNet have demonstrated impressive empirical advantages across diverse tasks compared to their more general tensor-product-based counterparts (e.g., TFN, Equiformer).
>
> Following the reviewer’s valuable guidance, we will revise our manuscript to briefly acknowledge the proven speed and empirical effectiveness of spherical-scalarization methods. Additionally, inspired by the insightful suggestions drawn from D-spanning [A], we will clarify that while scalarization methods offer substantial efficiency in specific applications, the tensor product framework maintains broader theoretical completeness. Thus, accelerating full tensor product computations—as achieved through our proposed Wigner 6j convolution—remains an important and relevant research direction.
>
> We greatly appreciate this thoughtful recommendation, as it provides us with a clearer path to accurately contextualize and strengthen the presentation of our contributions.

---

> > ### Comment · Reviewer_7r4S · 2025-08-05
> >
> > The authors’ response is satisfying. They have finally addressed the paper’s main challenge of **having both**, and the two key contributions are now well aligned.
> >
> > Additionally, I would like to offer a few minor suggestions, though there is no need for the authors to respond immediately:
> >
> > - How might the name "Wigner 6j conv" be made more accessible for wider dissemination? Would it be possible to shorten it further (at least by removing spaces), or perhaps include the name in the paper title to facilitate easier searching?
> > - Concerning the observation that the model performs less well than GotenNet on QM9 and MD17, I recommend including comparisons to HEGNN and SO3KRATES, as both methods utilize "spherical scalarization." For reference, in results reported in [B], E2Former could achieve superior results on LUMO and HOMO. A head-to-head comparison with HEGNN could further highlight the benefits of employing a similar transformer-like architecture.
> >
> > [B] Liu W, Zhu Y, Zha Y, et al. Rotation- and Permutation-equivariant Quantum Graph Neural Network for 3D Graph Data[J]. IEEE Transactions on Pattern Analysis and Machine Intelligence, 2025.

---

> ### Comment · Reviewer_7r4S · 2025-08-05
>
> I've raised my rating to Strong Accept and increased my confidence level to 5. The authors deserve it, enjoy it.

---

### Official Review · Reviewer_ejRh · 2025-06-13

**Clarity:** 3
**Significance:** 4
**Originality:** 4
**Rating:** 6
**Confidence:** 4

**Summary:**

This paper introduces E2Former, an SO(3)-equivariant transformer architecture designed for molecular modelling. The primary contribution is a new method which the authors call "Wigner 6j convolution", a reformulation of the computationally expensive tensor products that are central to equivariant GNNs based on SO(3) convolutions. By leveraging a binomial expansion of spherical harmonics and the Wigner 6j recoupling of angular momenta, the main computational burden is shifted from being dependent on the number of edges $\\lvert \\mathcal{E} \\rvert$ to being dependent on the number of nodes $\\lvert \\mathcal{V} \\rvert$. Reducing the complexity of the core tensor product operation from $\\mathcal{O}(\\lvert \\mathcal{E} \\rvert)$ to $\\mathcal{O}(\\lvert \\mathcal{V} \\rvert)$ results in a large computational efficiency gain, because $\\lvert \\mathcal{V} \\rvert$ is typically much smaller than $\\lvert \\mathcal{E} \\rvert$ (even in sparse graphs). The authors build the E2Former architecture around this efficient convolution and demonstrate its effectiveness through extensive experiments on various benchmarks, including OC20, OC22, and SPICE. The results show that E2Former achieves accuracy competitive with or superior to state-of-the-art models while being significantly faster in terms of both training and inference time. The practical utility of the model is further demonstrated through a stable, long-time molecular dynamics simulation that accurately reproduces DFT-level vibrational spectra.

**Questions:**

1. Can the authors please comment on the point outlined in Weakness 1? In particular, if I am mistaken, I'd appreciate a detailed explanation of what I am missing. If I am right about this, I hope the authors agree with me that this error should be fixed in the manuscript.

2. The paper cites "Euclidean Fast Attention" (Frank et al., 2024), which also achieves linear scaling for equivariant attention. Could you elaborate on the conceptual differences between the Wigner 6j convolution and the approach in that work? My understanding is that your method provides a general factorization of the tensor product itself, which could be used in any context (e.g., in a MACE-like model without attention), whereas their method is specific to linearizing the dot-product attention mechanism. Is this interpretation correct? A brief comparison in the related work section could be beneficial.

**Ethical Concerns:**

["NO or VERY MINOR ethics concerns only"]

**Final Justification:**

I raised my score to 6, because the authors addressed all weaknesses in the original manuscript.

The theoretical contribution of this paper will be very impactful for the subfield of equivariant neural networks based around irreducible representation of SO(3)/O(3).

**Limitations:**

yes

**Quality:**

3

**Strengths And Weaknesses:**

**Strengths**
1. The paper tackles a widely recognised bottleneck in the field of equivariant models for molecular systems, namely the computational cost of SO(3) tensor products. By providing a principled method to achieve $\\mathcal{O}(\\lvert \\mathcal{V} \\rvert)$ scaling, this work makes the core operation of many existing (and future) architectures relying on SO(3) convolutions more efficient.
2. The core technical contribution (Wigner 6j convolution) is highly original. While the mathematical tools used (Wigner 6j symbols and angular momentum theory) are well-established, their application to reformulate and accelerate equivariant convolutions in a GNN is novel and insightful. Leveraging a "binomial local expansion" combined with the recoupling scheme is a clever and powerful idea and an elegant mathematical reformulation.
3. The paper is well-written and organised. Despite the mathematical density of the topic, the authors do an excellent job of building intuition. Figure 1 provides a clear visual overview of the core idea before diving into the formal mathematics. The main text provides proof sketches that convey the key steps, while the appendix contains the full details.

**Weaknesses**
1. I might be wrong about this (in which case I apologise!), but I believe there is a rather large mistake in the manuscript (which, however, should be relatively easy to fix). The definition of spherical harmonics $Y^{(\\ell)}$ (Definition 2.1) seems correct to me, but the construction used to obtain the Wigner 6j convolution relies on a clever trick which seems incompatible with spherical harmonics. Rather, it seems that solid harmonics $R^{(\\ell)} = \\lVert \\vec{r} \\rVert^{\\ell} Y^{(\\ell)}$ are required to make it work. Namely, the identity $Y^{(1)}(\\vec{r}_{ij})=Y^{(1)}(\\vec{r}_i)-Y^{(1)}(\\vec{r}_j)$ (line 122) is crucial for the derivation, but it does not hold for spherical harmonics (it holds only for solid harmonics). The spherical harmonics $Y^{(\\ell)}_m$ (using the notation from the paper) for degree 1 are given by (using Definition 2.1 with Racah normalisation): $Y^{(1)}_1 = \\frac{x}{r}$, $Y^{(1)}\_{-1} = \\frac{y}{r}$, and $Y^{(1)}_0 = \\frac{z}{r}$, where $r=\\sqrt{x^2 + y^2 + z^2}$. It is easy to see that the required linearity condition does not hold for these. However, it *does* hold for the solid harmonics $R^{(1)}_1 = x$, $R^{(1)}\_{-1} = y$, and $R^{(1)}_0 = z$. In fact, the polynomials given for $Y^{(2)}$ in the text (lines 70 and 71) are *actually* expressions for $R^{(2)}$, because they lack the normalisation by $r^2$. I therefore believe that the authors have (unknowingly?) used solid harmonics for their implementation already (otherwise it would likely not work). The empirical results are thus (very likely) unaffected, but the authors should correct the description of their method in the revised manuscript. This also has some practical consequences for architecture design, which the authors may want to discuss, so that readers that want to use Wigner 6j convolutions are aware of them: Typical SO(3) convolutions use "filters" of the general form $g\_{\\ell}(r)Y^{(\\ell)}(\\vec{r})$, where $g\_{\\ell}(r)$ are (learnable) radial functions for the different degrees $\\ell$. The fact that Wigner 6j convolutions must instead use filters of the form $g\_{\\ell}(r)R^{(\\ell)}(\\vec{r})$  is important in so far that the "weight" of "interactions" between nodes implicitly increases by a factor $r^{\\ell}$, which is undesirable for typical applications (usually, the influence of distant nodes should *decrease*). Fortunately, this should not be an issue in practice, because it can be compensated for by scaling down the radial components of filters accordingly (i.e. $g\_{\\ell}(r)Y^{(\\ell)}(\\vec{r})=r^{-\\ell}g\_{\\ell}(r)R^{(\\ell)}(\\vec{r})$),  but readers should probably be made aware of this fact.
2. Typically, SO(3) convolutions use filters of the form $g\_{\\ell}(r)Y^{(\\ell)}(\\vec{r})=r^{-\\ell}g\_{\\ell}(r)R^{(\\ell)}(\\vec{r})$ (see above), because it makes the operation much more expressive (the radial component allows to discriminate between nodes based on distance). When this construction is used, then the whole Wigner 6j convolution *technically* still scales $\\mathcal{O}(\\lvert \\mathcal{E} \\rvert)$, because in general, $g\_{\\ell}(r)$ needs to be evaluated for all $r_{ij}$. Of course, the cost of the expensive tensor product is still greatly reduced (so a large speedup is to be expected anyway), but this detail should be discussed in the text. Perhaps it would be more precise to state that only the tensor product component of the convolution is reduced to $\\mathcal{O}(\\lvert \\mathcal{V} \\rvert)$.
3. The methodology is heavily reliant on advanced concepts from representation theory and quantum angular momentum theory (irreps, Clebsch-Gordan coefficients, Wigner 3j/6j symbols). This high barrier to entry may limit the immediate understanding and adoption of the method by members of the broader NeurIPS community who are not specialised in this subfield. This is more of an inherent characteristic of the topic than a flaw in the paper's execution, but it should still be considered. While the authors provide code for their implementation, I imagine an additional standalone script that exemplifies the core idea on a simple "toy task" (perhaps showing numerically that Wigner 6j convolution and ordinary SO(3) convolution are equivalent for a small point cloud) would be helpful to many readers. This way, many advanced concepts could be "demystified", for example, solid harmonics are just very simple polynomials in Cartesian coordinates, Clebsch-Cordan coefficients are just a tensor of certain numerical constants etc.

---

> ### Author Rebuttal · Authors · 2025-07-28
>
> Thank you for your detailed review and helpful feedback. We address your concerns below and describe the improvements we will make in our revision.
>
> **W1 - 2**  We sincerely thank the reviewer for their exceptionally thoughtful and precise observation. The reviewer is correct to highlight the distinction between spherical and solid harmonics, and we confirm that our model does, in fact, use solid harmonics.
>
> Our implementation realizes this by computing the normalization factor $\|r\|^l$ externally and applying it to the spherical harmonic basis functions $Y^l(\hat{r})$. The empirical correctness of our model, which the reviewer correctly inferred, stems from this choice.
>
> We agree this is an important detail that was insufficiently clarified. We will revise the manuscript to state our use of solid harmonics explicitly. Furthermore, we will add a clarification on the design of the filter functions, noting the need to adjust the radial component via scaling ($r^{-l}g_l(r)$) to counteract the implicit weighting from the solid harmonic basis.
>
> We are grateful for this feedback, which significantly sharpens both the theoretical exposition and practical guidance of our paper.
>
> **W3** We sincerely thank the reviewer for their insightful and constructive comments. The reviewer has astutely highlighted the accessibility challenges associated with the advanced mathematical formalism in our paper. While these concepts are integral to the rapidly evolving field of equivariant deep learning at NeurIPS, we fully agree that enhancing accessibility is essential to maximize the impact of our work.
>
> In response to this valuable feedback, we propose the following steps to improve clarity and accessibility:
>
> First, as suggested, we will develop a standalone tutorial script. This script will demonstrate a practical “toy task,” clearly illustrating the mechanics of our method and numerically validating our theoretical claims.
>
> Second, following the reviewer’s advice, we will introduce a new appendix section explicitly connecting abstract concepts—such as solid harmonics and Clebsch-Gordan coefficients—to straightforward polynomial functions and tangible numerical tensors. For example, we will explicitly present the solid harmonics for $l=1$ and $l=2$ in their Cartesian polynomial forms to effectively bridge abstract mathematical concepts to familiar, concrete functions. Additionally, we will enhance the clarity of these concepts in the main text itself.
>
> We believe this  approach will improve the transition from theory to practical implementation. We thank the reviewer once again for this excellent guidance, which will substantially enhance both the impact and reach of our work.
>
> **Q2** Thank you for your insightful observation regarding the relationship between our Wigner 6j convolution approach and the Euclidean Fast Attention method by Frank et al. (2024). Your understanding is exactly correct!
>
> The fundamental distinction lies in the scope and target of optimization. Our method provides a general factorization framework for the tensor product operation itself, which constitutes the computational bottleneck in equivariant neural networks. This factorization is agnostic to the specific architecture in which it is deployed—it can accelerate tensor products whether they appear in attention mechanisms, message-passing layers, or other equivariant operations such as those in MACE-like models. In contrast, the Euclidean Fast Attention method specifically targets the linearization of the dot-product attention mechanism through kernel approximations.
>
> These two acceleration strategies operate at different levels of the computational hierarchy and are indeed orthogonal to each other. The Euclidean Fast Attention reduces the quadratic complexity of attention mechanisms to linear complexity in sequence length, while our Wigner 6j convolution reduces the complexity of the underlying tensor product operations that appear within each attention computation (or any other equivariant operation). Therefore, the two methods can be naturally combined: one could apply Euclidean Fast Attention to achieve linear scaling in sequence length while simultaneously using our Wigner 6j factorization to accelerate the tensor products within the attention computation.
>
> We recognize the potential for such combinations and have explicitly identified this as a promising direction in our future work section. The integration of both techniques could potentially yield multiplicative performance improvements, particularly for large-scale equivariant models that employ attention mechanisms. We will clarify this distinction and the complementary nature of these approaches in our revised related work section to better position our contribution within the broader landscape of acceleration methods for equivariant neural networks.

---

> > ### Comment · Reviewer_ejRh · 2025-08-01
> >
> > I thank the authors for their detailed reply and clarifications on the points I raised in my review. I believe the changes the authors want to include in the camera-ready version will greatly enhance the manuscript and the tutorial/example script the authors will provide will make the proposed method much more accessible to readers.
> >
> > In light of the other reviews (who gave this paper a much lower average score than seems appropriate in my view), I'd like to reiterate that this paper presents a great methodological contribution, which can potentially increase the efficiency of many equivariant models based around irreducible representations of SO(3)/O(3). The proposed method is very general and can be made mathematically equivalent to a lot of existing "message-passing-like" update functions in other models with very small tweaks, so it enables a sort of "plug-in" speedup. While the implementation details may be a bit complicated, this will be mitigated by the authors providing example scripts, and I expect this method to be adopted by essentially all model architectures that use compatible operations in the future (it's comparable to the FlashAttention algorithm for attention-based models, it just targets a more specialised kind of architecture). I have the impression that most of the other reviewers do not appreciate this enough.
> >
> > It would be a shame if this paper was rejected form NeurIPS for silly reasons like the proposed E2Former architecture not achieving the absolute best result on all benchmarks. The E2Former architecture just demonstrates empirically that the proposed method works in practice, but the methodological and theoretical contribution of this paper is far more important.

---

> > > ### Comment · Reviewer_7r4S · 2025-08-01
> > >
> > > Thank you, Reviewer ejRh, for your enthusiastic and detailed feedback. I do have a question, though. As you noted, this method is intended to be a versatile, plug-and-play solution. In light of this, could you clarify what you see as the core connection between the authors’ method and the E2Former architecture? You praised the potential for broad efficiency gains in equivariant models based on irreducible representations of SO(3)/O(3), and I generally concur with this assessment. However, the paper currently lacks direct empirical evidence to support this claim—while the authors state that their approach is mathematically equivalent to existing alternatives, the only relevant placement appears to be Section 4.1, without any concrete numerical validation.
> > >
> > > In fact, it was not until the analysis provided in response to my W3.1 raised the issue that any experimental verification of this numerical equivalence was conducted—prior to that, the claims were purely theoretical. Furthermore, it was through my questioning that the observation emerged: the method introduces minor improvement, and I guass it may enhance model training stability.
> > >
> > > Setting this aside, I would like to highlight an even bigger inconsistency: if operations like Wigner 6j are truly mathematically equivalent, why does the proposed model still underperform compared to the baseline? The architectural contributions you emphasized in your latest response, in my view, are not substantiated by the content of this manuscript. On the contrary, if the operations are indeed mathematically equivalent as claimed, doesn't the fact that E2former fails to match state-of-the-art performance actually suggest that there are shortcomings in the architecture—even if, of course, there are also certain advantages?
> > >
> > > My broader concern is that the paper lacks a coherent, unified perspective. Solving large-scale atomic modeling tasks fundamentally requires both computational speedup and scalability (or other contribution): Wigner 6j addresses the former, while E2Former architecture might address the latter. If, as you suggest, we only focus on Wigner 6j, we risk overlooking essential aspects of the work. Thus, we need the authors to clearly articulate the architectural contributions—this needs to be backed by rigorous experimental evidence, not left to assumption or extrapolation.
> > >
> > > Finally, let us strive for an objective and balanced evaluation of the work itself, rather than unduly influencing other reviewers (including ACs or SACs) through disproportionately high ratings that dismiss legitimate criticisms. I am sure your support for the authors comes from a positive place, but I would caution that ungrounded enthusiasm can, if anything, diminish the overall perception of the manuscript. Let’s keep our discussion focused on the evidence provided.

---

> > > > ### Comment · Reviewer_ejRh · 2025-08-01
> > > > **Response to reviewer 7r4S**
> > > >
> > > > I definitely see the proposed Wigner 6j coupling as the core contribution of the paper (which would even be publishable without any empirical results in my opinion). The E2Former architecture is merely a nice application of this method, but I see this as more of an afterthought. New architectures tend to become outdated within 3 months anyway, and I find the field's increasingly strong focus on "SOTA results" on benchmarks a worrying development. Important theoretical and methodological insights unfortunately tend to be dismissed and overlooked (which is probably also why the authors put more focus on the architecture than the methodological aspects in their manuscript).
> > > >
> > > > It is obvious from well-known representation theory that the method presented by the paper is mathematically equivalent to the standard implementation (although some important details need to be considered, see for example the comment on radial functions in my review). The original manuscript definitely was imprecise in some important technical aspects (e.g., solid harmonics vs. spherical harmonics), but the authors acknowledged these flaws and agreed to fix them. Also, note that the authors did not invent Wigner 6j recoupling itself (which has been a standard method to analyse angular momentum in quantum mechanics since roughly a century, so the equivalence is well-known), but coming up with the idea to use this as a "mathematical trick" to shift the computational cost of tensor operations from edge-based to node-based is still non-trivial (otherwise the idea would be well-known and widely used already). Nonetheless, I understand that not all readers are familiar enough with the relevant mathematics to see the equivalence without empirical evidence, which is also why I recommended that the authors include a small reference script to show this (and generally try to make the method more widely accessible).
> > > >
> > > > I have read the points about the proposed architecture underperforming certain baselines, but I don't see how this is related to the mathematical equivalence of the tensor operation. There are many possible design choices that go into an architecture and influence its performance, and indeed there may be shortcomings here. However, the mathematical equivalence is much better shown empirically with a standalone script that focuses on just the operation in question and isolates it from other design choices (which the authors agreed to do). I don't see why underperformance of the E2Former on certain benchmarks should be considered an "inconsistency" in this context.
> > > >
> > > > With regards to the last paragraph in your comment, I do not appreciate that you suggest my evaluation is not objective or  balanced, nor that you call my enthusiasm about this work "ungrounded", or suggest that I am "unduly influencing other reviewers (including ACs or SACs)" or "dismissing legitimate criticisms". Opinions may (obviously) differ, but I am certain that other reviewers (including ACs or SACs) are able to form their opinion without letting my statements influence them. Finally, while I am admittedly quite surprised by the low average ratings of the other reviewers (and especially by your recommendation to reject the paper!), I want to emphasise that I do not consider my rating "disproportionately high". In my opinion, it is an adequate rating for a paper with contributions that are likely to be very impactful.

---

> > > > > ### Comment · Reviewer_7r4S · 2025-08-01
> > > > >
> > > > > I understand your point—the innovation brought by simply accelerating with Wigner 6j is indeed quite interesting. Of course, we can further discuss what additional contributions or evidence might be needed.
> > > > >
> > > > > Regarding mathematical equivalence, my view is that, since the Wigner 6j acceleration is mathematically equivalent to the original tensor product, whether or not Wigner 6j acceleration is applied should not significantly change the results of E2Former. This means that if the E2Former with Wigner 6j acceleration underperforms compared to other baselines, then so would the E2Former without Wigner 6j acceleration. Conversely, if it outperforms the baselines, this would only indicate that the original E2Former itself is strong, and not necessarily showcase the specific advantage of Wigner 6j acceleration. In other words, this architectural change does not provide meaningful evidence to support the merits of Wigner 6j acceleration.
> > > > >
> > > > > In your initial reply, you also mentioned the "plug-in" nature and pointed out that "The E2Former architecture just demonstrates empirically that the proposed method works in practice, but the methodological and theoretical contribution of this paper is far more important." I completely agree with this view. My current concern is that, if we only look at the original experiments (ignoring the ablations I suggested the authors add), how will others interpret the role of Wigner 6j acceleration? When all modules are coupled together, it is difficult to tell whether Wigner 6j acceleration is responsible for E2Former’s underperformance, or if there are other reasons. Such a line of argument is not sufficiently justified. Therefore, in my second response, I requested the authors to apply Wigner 6j acceleration to common backbones such as TFN, MACE, and Equiformer. This would demonstrate that Wigner 6j acceleration is “plug-and-play,” and would further allow exploration of its potential effects on training stability. I hope this clarifies my reasoning—that is, the theoretical claims of the authors do not align well with their experimental evidence.
> > > > >
> > > > > Additionally, focusing solely on Wigner 6j acceleration would inevitably require the authors to remove discussion of the model architecture. However, it is important to note that we are reviewing for NeurIPS rather than ICLR, and the authors do not have the opportunity to upload a revised PDF. This could potentially lead to the paper being recommended for transfer due to “substantial changes,” which might not be in the authors' best interest. Therefore, I suggest that the authors frame their contributions with a broader vision that connects both aspects, minimizing the need for major revisions and increasing the chance of acceptance.

---

> > > > > > ### Comment · Reviewer_ejRh · 2025-08-01
> > > > > >
> > > > > > I am glad to hear that you generally agree that the Wigner 6j coupling is interesting. Also, I agree that applying the Wigner 6j trick to established architectural backbones would provide strong empirical evidence. However, I also recognise that modifying existing architectures may constitute a substantial amount of work (e.g., because of the authors' unfamiliarity with the respective codebases etc.) that cannot necessarily be performed within the limited time frame the authors have at their disposal. Therefore, I suggested a standalone script to show the equivalence empirically, which can be written much more quickly. I believe this would provide very similar evidence compared to the experiment you suggest. Why do you believe changing how a particular operation is implemented would influence training stability? I can only see this happening because of numerical precision issues, which should already become evident in a "standalone" comparison (if they exist).
> > > > > >
> > > > > > Perhaps I was unclear about this, but I did not intend to suggest the authors should remove the discussion of model architecture. I merely wanted to express my opinion that I find the theoretical contribution significant enough to merit publication, even if there weren't any empirical studies.

---

> > > > > > > ### Comment · Reviewer_7r4S · 2025-08-01
> > > > > > >
> > > > > > > I think this makes sense, and it certainly seems like a reasonable approach. Since NeurIPS does not allow code submission, I suggest reporting the mean and variance of $\\|\texttt{model\\_w/\\_Wigner-6j}(\mathcal{G})-\texttt{model\\_w/o\\_Wigner-6j}(\mathcal{G})\\|$ as an alternative. Notably, the model does not need to be trained for this analysis. However, I would still encourage the authors to carefully consider how to present the relationship between the two main innovations in the paper. Thoughtful integration is important to avoid any potential risk of rejection due to "major changes."
> > > > > > >
> > > > > > > Regarding my speculation about improved training stability—admittedly, this is just a hypothesis. My intuition comes from recently working with a large equivariant model (>1B parameters), where numerical instability was a significant challenge. The authors’ approach appears to have some promise in this regard, so I hope they can further explore whether it meaningfully addresses such issues. Of course, if it does not, that is completely understandable as well.
> > > > > > >
> > > > > > > There are also some minor pieces of evidence. For example, the authors chose not to adopt my suggestion of initializing the coordinates to be decentralized—meaning there can be larg translation offsets. While these translations will eventually cancel out, repeated tensor products of intermediate values could lead to considerable numerical offsets. Interestingly, the model's results do not seem to be significantly affected in such cases, which I suspect may help with training stability.

---

> > > > > ### Comment · Reviewer_7r4S · 2025-08-01
> > > > >
> > > > > Furthermore, after reading your first paragraph, I wholeheartedly agree with your viewpoint and am genuinely touched by your perspective. I have great respect for colleagues like you who are dedicated to elevating the standards of our community. I have also found myself passionately advocating for papers I truly admired during discussions with those meaner man, sometimes giving them the highest possible scores. At the same time, I have experienced the disappointment when my own work was rejected by reviewers who valued SOTA results over theoretical contributions. While the pursuit of elegant theory remains a noble goal, for this paper, I believe it is crucial for authors to provide consistent and convincing evidence to support their claims. Otherwise, I feel it is preferable for an incomplete submission to be set aside, allowing the work to grow and mature before being considered again.

---

### Official Review · Reviewer_t8Wk · 2025-06-25

**Clarity:** 3
**Significance:** 3
**Originality:** 3
**Rating:** 5
**Confidence:** 3

**Summary:**

The paper introduces the E2Former, a new spherically equivariant architecture for molecular modeling. At its core the model uses the novel Wigner 6j convolution, which is mathematically equivalent to conventional SO(3) convolutions, but scales with the number of nodes instead of the number of edges, significantly reducing computational cost. E2Former shows competitive results on several benchmark datasets and is also demonstrated to perform well in molecular dynamics simulations.

**Questions:**

- At the bottom of page 5 I am not sure why the identity reads $A \otimes (B \otimes C) \cong (A \otimes B) \otimes^{6j} C$ shouldn’t the two sides be equal and not just isomorphic according to equation (2.4)?
In the following sentence the effective commutativity of the Clebsch-Gordan decomposition is invoked (leading to another isomorphism) but is this really needed here? Since the commutation happens inside a projector onto l-irreps, commutation should be possible without using effective commutativity.
As far as I understand the key point of the derivation is that the two sides in Theorem 3.3. are equal and not just isomorphic. So even though both isomorphisms are of course formally correct, I think it would add clarity to emphasize the equality of the expressions.
- Could the authors include an explicit reference to [1], the original paper of the SCN baseline, which is missing so far?
- Could the authors also provide results for their model on the OC22 S2EF-Total test split and include the results of eSCN in the table?

Minor comment:
- On page 4 in theorem 3.2., why is there a ‘k’ index on the spherical harmonic? It seems to me like it should be either an ‘l’ or a ‘u’

I am willing to increase my score if the above points are adequately addressed.

**Ethical Concerns:**

["NO or VERY MINOR ethics concerns only"]

**Final Justification:**

The rebuttal addressed my remaining concerns. I think that especially the introduction of the novel Wigner 6j convolution layer is is very relevant for the community, justifying the publication at NeurIPS.

**Limitations:**

The authors mention limitations in the NeurIPS checklist and state that they were also discussed in section 5. However this discussion is apparently no longer present in section 5. It would be good to reintroduce the discussion of limitations into the main text.

**Quality:**

3

**Strengths And Weaknesses:**

- Strengths:
    - The paper tackles a relevant problem, namely the acceleration of equivariant GNN architectures in order to scale them to increasingly large molecular systems. The introduced Wigner 6j convolution has the potential to be used to speed up a variety of different architectures.
    - The math behind the Wigner 6j convolution is explained in an intuitive fashion, while also being rigorously derived in the Appendix.
- Weaknesses
    - The discussion of baselines in the experiments section is incomplete. The SCN architecture for example is never mentioned in the text and the corresponding paper [1] is not cited.
    - The number of baselines for the OC22 dataset is very small. It is also not clear why the authors decide to only report metrics on the validation split. Evaluating the performance also on the test split would allow to also compare against the eSCN architecture (like in [2]).

[1] Zitnick, C. L et. al. Spherical channels for modeling atomic interactions. In Advances in Neural Information Processing Systems, 2022.

[2] Yi-Lun Liao et. al. Improved equivariant transformer for scaling to higher-degree representations. arxiv preprint arxiv:2306.12059, 419 2023.

---

> ### Author Rebuttal · Authors · 2025-07-28
>
> Thank you for your detailed review and helpful feedback. We address your concerns below and describe the improvements we will make in our revision.
>
> **W1-2 Q2-3** We agree with the reviewer that a more thorough comparison with relevant baselines strengthens the paper. We apologize for the omission of the **SCN** architecture [1] in our discussion and bibliography. In our revised manuscript, we have:
>
> 1. Added a proper citation and discussion of the **SCN** model in the related work and experiments sections.
> 2. Expanded our results on the Open Catalyst 2022 (OC22) dataset to include the **S2EF-Total test split**. This allows for a direct comparison with more models, including **eSCN**.
>
> The new results are presented in the table below and have been integrated into the main paper. **E2Former** demonstrates highly competitive performance, outperforming **eSCN** and **GemNet-OC** on energy predictions and achieving strong results on forces, all while maintaining its significant computational efficiency advantages.
>
> | Model | Energy MAE (meV) ID | Energy MAE (meV) OOD | Force MAE (eV/Å) ID | Force MAE (eV/Å) OOD |
> | --- | --- | --- | --- | --- |
> | **E2Former (ours)** | 345 | 745 | 0.0243 | 0.0352 |
> | eSCN | 350 | 789 | 0.0238 | 0.0357 |
> | EquiformerV2 | 263.7 | 659.8 | 0.0216 | 0.0327 |
> | GemNet-OC | 374 | 829 | 0.0294 | 0.0396 |
>
> **Q1** You are correct. Thank you for this insightful comment and for spotting this inconsistency.
>
> The relationship should indeed be a strict equality, not an isomorphism. The main point of our derivation, particularly for Theorem 3.3, is that the expressions are strictly equal, and using the isomorphism symbol $\cong$ is misleading and detracts from this key fact.
>
> We will revise the manuscript to change the expression on page 5 from $A \otimes (B \otimes C) \cong (A \otimes B) \otimes^{\mathrm{6j}} C$ to $A \otimes (B \otimes C) = (A \otimes B) \otimes^{\mathrm{6j}} C$. As you rightly pointed out, this equality follows directly, and the subsequent mention of "effective commutativity" is unnecessary in this context. We will remove this justification to avoid confusion and improve the clarity of the argument.
>
> **Minor comment** Thank you for identifying these notational inconsistencies. You are correct—the notation should be $l$. These typographical errors will be corrected promptly. We appreciate your thorough review and attention to detail.

---

> > ### Comment · Reviewer_t8Wk · 2025-08-01
> >
> > I thank the authors for their answers to my questions and the new experiments that were conducted.
> >
> > The rebuttal addressed most of my remaining concerns. I agree with reviewer 7r4S that the discussion of the architecture could be improved in particular with regard to how it compares to other SOTA baselines and I would encourage the authors to do so. However for me the main contribution of the paper is the introduction of the Wigner 6j convolution layers, which I think is also reflected in the amount of actual content devoted to that part. Given the mathematical equivalence to traditional CG convolutions (the discussion of which is further enhanced by the new experiments which were requested by other reviewers) I think it is not necessary to extend existing baselines using the method and I appreciate the discussion of design philosophies behind E2Former motivated by the underlying faster convolution layers. Overall I thus think that the paper introduces a very interesting method which might well find many practical applications. I have raised my score.

---

### Official Review · Reviewer_ZFTk · 2025-07-01

**Clarity:** 2
**Significance:** 2
**Originality:** 2
**Rating:** 4
**Confidence:** 2

**Summary:**

This paper introduces a novel graph neural network architecture designed to address computational inefficiencies in Equivariant Graph Neural Networks (EGNNs), particularly for applications in molecular modeling. The key innovation is the introduction the concept of Wigner 6j coupling from physics, which shifts the computational focus from edges to nodes in molecular graphs, reducing the complexity from edge-dependent $O(|E|)$ to node-dependent $O(|V|)$.

**Questions:**

Please consult the Weaknesses for questions.

**Ethical Concerns:**

["NO or VERY MINOR ethics concerns only"]

**Final Justification:**

I'm generally satisfied with the response of my review and also other reviewer's reviews. Although the practical usage of E2Former still remains, I will raise my score to the positive side of the borderline.I'm generally satisfied with the response. Although the practical usage of E2Former still remains, I will raise my score to the positive side of the borderline.

**Limitations:**

yes

**Paper Formatting Concerns:**

No major formatting issues.

**Quality:**

2

**Strengths And Weaknesses:**

Strengths:

1.  Methodology: The introduction of Wigner 6j convolution is new for the graph learning community, it shifts computation from edge-based to node-based operations, which is beneficial for handling large molecular systems efficiently.

2. The proposed E2Former shows good experimental results on OC20 and SPICE dataset.

Discussion on Expressiveness:
The paper lacks a thorough discussion regarding the expressiveness of the Wigner 6j convolution. It remains unclear whether the convolution method proposed in E2Former retains the full expressive power compared to traditional full tensor product methods.
Motivation in Context:

Given the extensive research focused on accelerating Clebsch-Gordan (CG) tensor products of spherical harmonics, the motivation for introducing E2Former may appear somewhat limited. More clarity on how this approach distinctly advances existing acceleration techniques would be beneficial.

Organizational Structure:
The paper’s organization can be improved for clarity. For instance, the precise definition and implications of "linear-scaling" should be prominently highlighted in the introduction to better guide readers through the significance and impact of the proposed approach.

Missing related works on invariant neural networks with invariant polynomials:
[1] Weisfeiler Leman for Euclidean Equivariant Machine Learning;   ICML 2024
[2] A new perspective on building efficient and expressive 3D equivariant graph neural networks;    Neurips 2024

---

> ### Author Rebuttal · Authors · 2025-07-28
>
> Thank you for your detailed review and helpful feedback. We address your concerns below and describe the improvements we will make in our revision.
>
> **W.1** The reviewer raised an important and insightful question regarding the clarity of expressiveness of the Wigner 6j convolution. We deeply appreciate this thoughtful observation and acknowledge that this fundamental aspect warrants thorough clarification. To address this valuable concern, we emphasize that our Wigner 6j convolution maintains the exact expressive power equivalent to conventional SO(3) convolution methods.
>
> **Theoretical Equivalence**: As demonstrated in Theorems 3.2 and 3.3, the Wigner 6j convolution provides exact representation of the complete angular basis used by traditional CG-based convolutions. The methods are therefore mathematically equivalent by construction. The Wigner 6j recoupling represents a unitary transformation of angular momentum channels, ensuring no loss in representational capability while delivering significant computational benefits.
>
> **Empirical Validation**: The results presented in Figure 2(b–d) provide evidence of this theoretical equivalence, demonstrating that our convolution produces numerically **identical outputs** compared to conventional SO(3) convolution. This empirical validation confirms the expressiveness equivalence while achieving significant efficiency improvements that make the approach practically advantageous.
>
> Following the reviewer's feedback, we will add a dedicated subsection to the main text that directly addresses expressiveness equivalence. This addition will clarify the theoretical foundation and demonstrate that representational power is fully preserved.
>
> **W.2** We greatly appreciate the reviewer's insightful perspective on existing CG acceleration methods. Their observation highlights an important distinction that underscores the unique value of our contribution.
>
>
> Prior methods such as eSCN (ICML ’23), Gaunt Tensor Product (ICLR ’24), and SPHNet (ICML ’25) optimize individual tensor product computations, significantly accelerating performance yet remaining constrained by $O(|\mathcal{E}|)$ scaling, as each edge still necessitates a separate tensor product calculation. E2Former introduces a paradigm shift by drastically reducing the total number of tensor products. Leveraging Wigner 6j recoupling theory, we transform computations from an edge-based approach to a node-based formulation, reducing complexity from $O(|\mathcal{E}|)$ to $O(|\mathcal{V}|)$. Importantly, our method is orthogonal to and complements existing Clebsch–Gordan (CG) optimizations; the smaller set of node-based tensor products can further exploit these optimized implementations, offering compounded efficiency gains. We will explicitly highlight this synergistic potential as part of our future work.
>
>
> **W.3** We appreciate the reviewer's valuable feedback regarding the need for clearer organization and more prominent definition of "linear scaling" in the introduction. To address this concern, we will add an explicit definition of "linear scaling" at the beginning of the introduction, clarifying that this refers to computational complexity that scales linearly with respect to the number of tensor products, which is $O(\|\mathcal{V}\|)$ where $\|\mathcal{V}\|$ represents the number of nodes. We apologize that this fundamental distinction was not made sufficiently clear in the original manuscript.
>
> **W.4** We appreciate the reviewer's suggestion to include recent works on invariant polynomial networks, specifically "Weisfeiler Leman for Euclidean Equivariant Machine Learning" (ICML 2024) and "A New Perspective on Building Efficient and Expressive 3D Equivariant Graph Neural Networks" (NeurIPS 2024). These works offer complementary perspectives through different invariance structures and polynomial expansions. We will incorporate discussion of these works in Section 6 (Related Work) and identify their integration with our node-centric convolution as a promising future research direction.

---

> ### Comment · Area_Chair_Tab5 · 2025-08-06
>
> Dear reviewer,
>
> It seems you decided to increase your score to 4 but forgot to do so.
>
> AC

---

### Decision · Program_Chairs · 2025-09-17

**Decision:**

Accept (spotlight)

**Comment:**

This paper introduces a method to accelerate Clebsch Gordan computations in SO(3) networks via Wigner 6j convolutions. The reviewers agreed on the merit and timeliness of the approach, but initially voiced some concerns regarding technical details, related work and experiments. All these concerns were addressed in the rebuttal and the reviewers anonymously are very supportive of the paper. The reviewers now feel that this paper consists of an important and general contribution which can speed up the many architectures for 3D data which are based on 3D convolutions in Fourier space. Thus I strongly recommend acceptance as a spotlight presentation.